# Atg43 tethers isolation membranes to mitochondria to promote starvation-induced mitophagy in fission yeast

**Tomoyuki Fukuda[1]\*, Yuki Ebi[1], Tetsu Saigusa[1], Kentaro Furukawa[1], Shun-ichi Yamashita[1], Keiichi Inoue[1], Daiki Kobayashi[2], Yutaka Yoshida[3], Tomotake Kanki[1]\***

[1]Department of Cellular Physiology, Niigata University Graduate School of Medical and Dental Sciences, Niigata, Japan; [2]Omics Unit, Niigata University Graduate School of Medical and Dental Sciences, Niigata, Japan; [3]Department of Structural Pathology, Kidney Research Center, Niigata University Graduate School of Medical and Dental Sciences, Niigata, Japan

**Abstract** Degradation of mitochondria through mitophagy contributes to the maintenance of mitochondrial function. In this study, we identified that Atg43, a mitochondrial outer membrane protein, serves as a mitophagy receptor in the model organism *Schizosaccharomyces pombe* to promote the selective degradation of mitochondria. Atg43 contains an Atg8-family-interacting motif essential for mitophagy. Forced recruitment of Atg8 to mitochondria restores mitophagy in Atg43-deficient cells, suggesting that Atg43 tethers expanding isolation membranes to mitochondria. We found that the mitochondrial import factors, including the Mim1–Mim2 complex and Tom70, are crucial for mitophagy. Artificial mitochondrial loading of Atg43 bypasses the requirement of the import factors, suggesting that they contribute to mitophagy through Atg43. Atg43 not only maintains growth ability during starvation but also facilitates vegetative growth through its mitophagy-independent function. Thus, Atg43 is a useful model to study the mechanism and physiological roles, as well as the origin and evolution, of mitophagy in eukaryotes.

**\*For correspondence:**
tfukuda@med.niigata-u.ac.jp (TF);
kanki@med.niigata-u.ac.jp (TK)

**Competing interests:** The authors declare that no competing interests exist.

## Introduction

Mitochondria play crucial roles in a variety of cellular processes, including energy production, calcium homeostasis, metabolism, cell signaling, and apoptosis (*Rizzuto et al., 2012*). Mitophagy, the selective removal of mitochondria through autophagic degradation, is thought to contribute to mitochondrial quality and quantity control by eliminating dysfunctional or excessive mitochondria (*Fukuda and Kanki, 2018*; *Pickles et al., 2018*). Autophagy is a catabolic process that is highly conserved from yeast to humans. In autophagy, the cytoplasmic constituents are sequestered by the isolation membrane leading to the formation of a double-membrane vesicle, which is termed the autophagosome. The autophagosome subsequently fuses with the lysosome/vacuole leading to the degradation of its contents. Currently, over 40 autophagy-related (Atg) proteins, which are involved in different processes of autophagy, have been identified and characterized, mainly in the budding yeast *Saccharomyces cerevisiae*, and many have orthologs in other eukaryotes including mammals (*Mizushima, 2018*; *Nakatogawa et al., 2009*).

Multiple selective autophagy pathways eliminate particular cargo, including organelles, protein aggregates, and invading bacteria (*Gatica et al., 2018*). In selective autophagy, the cargo is recognized via specific receptor proteins present on the surface of the cargo. Such receptors contain the Atg8-family-interacting motif (AIM) or the LC3-interacting region (LIR), which is a highly conserved

sequence that interacts with Atg8 or its mammalian counterparts, such as LC3s and GABARAPs, which localize to the isolation membrane (*Gatica et al., 2018*; *Noda et al., 2010*). In mammals, multiple mitochondrial outer membrane (MOM) proteins, including NIX, BNIP3, FUNDC1, BCL2L13, and FKBP8, have been reported to serve as receptors that promote mitophagy through interactions with LC3s/GABARAPs via the LIR motif (*Kirkin and Rogov, 2019*). Besides receptor-mediated mitophagy, there is a mitochondrial ubiquitination-dependent pathway, in which LIR-containing ubiquitin-binding adaptors, such as OPTN and NDP52, link LC3s/GABARAPs to damaged mitochondria where MOM proteins are ubiquitylated by Parkin (*Lazarou et al., 2015*). NDP52 interacts with the core autophagy protein FIP200; this interaction is crucial for the targeting of the autophagy machinery to the cargo (*Vargas et al., 2019*). In budding yeast, mitophagy is dependent on Atg32, a receptor protein that is anchored to the MOM (*Kanki et al., 2009*; *Okamoto et al., 2009*). Upon induction of mitophagy, casein kinase two phosphorylates Atg32, thereby enabling its interaction with Atg11, an ortholog of FIP200 (*Aoki et al., 2011*; *Kanki et al., 2013*). Atg32 also binds to Atg8 via an AIM. However, the AIM plays only a limited role in mitophagy, whereas the interaction between Atg32 and Atg11 is essential (*Aoki et al., 2011*; *Farré et al., 2013*; *Kanki et al., 2013*; *Kondo-Okamoto et al., 2012*). Therefore, mitochondria are recognized as cargo for selective autophagy by similar but different mechanisms.

In the fission yeast *Schizosaccharomyces pombe*, a model organism that is evolutionarily distant from *S. cerevisiae*, autophagic degradation of mitochondria has been shown to be induced by nitrogen starvation (*Takeda et al., 2010*; *Zhao et al., 2016*). However, the molecular processes underlying the degradation of mitochondria, including whether this process is selective to mitochondria, remains unclear. In this study, we identified that the fission yeast MOM protein Atg43 serves as a mitophagy receptor. We found that Atg43 tethers Atg8 to mitochondria via an AIM to promote selective autophagy, similarly to mammalian mitophagy receptors. Atg43 also has a mitophagy-independent cellular function that facilitates vegetative growth, suggesting that Atg43 acquired the autophagic function via convergent evolution. Our findings shed light on molecular and evolutionary aspects of mitophagy in eukaryotes.

## Results

### Identification of genes required for starvation-induced degradation of mitochondria in fission yeast

To investigate autophagic degradation of mitochondria in fission yeast, we used C-terminal mRFP-tagged Tuf1, which is a protein in the mitochondrial matrix (*Chiron et al., 2005*). As fluorescent proteins, such as RFP and GFP, are relatively resistant to vacuolar proteases, autophagic degradation can be assayed by the production of free fluorescent proteins that are processed from their tagged proteins (*Meiling-Wesse et al., 2002*). In cells expressing Tuf1-mRFP, free mRFP was observed during nitrogen starvation (*Figure 1—figure supplement 1A*). The production of free mRFP was dependent on the core autophagy protein Atg1 and the vacuolar protease Isp6 (*Figure 1—figure supplement 1A*), confirming that Tuf1 was translocated to the vacuole and degraded by autophagy.

Fission yeast does not contain any proteins that are orthologous to budding yeast Atg32 or mammalian mitophagy receptors. The Fun14 protein coded by SPAC29A4.17c belongs to the Fun14 family, which includes the mammalian mitophagy receptor FUNDC1. However, Fun14 was found to be dispensable for mitochondrial degradation by autophagy (*Figure 1—figure supplement 1B*). To identify genes involved in autophagic degradation of mitochondria in fission yeast, we used the Bioneer knockout library (*Kim et al., 2010*) and screened for mutants defective in Tuf1-mRFP processing. Currently, we obtained two such mutants that have not been reported to be involved in autophagy-related processes. One mutant has a deletion in the gene of unknown function, SPAC14C4.01c, and the other in the *tom70*[+] gene, which encodes a highly conserved protein that is involved in mitochondrial protein import (*Baker et al., 2007*). We named the SPAC14C4.01c gene *atg43* according to the unified nomenclature for autophagy-related genes. Processing of Tuf1-mRFP was completely defective in the *atg43* deletion mutant, whereas a slight signal of free mRFP was detected in the *tom70* deletion mutant (*Figure 1A*). The deletion allele of *atg43*[+] (named *atg43-1*) was PCR-amplified from the Bioneer strain and transferred to

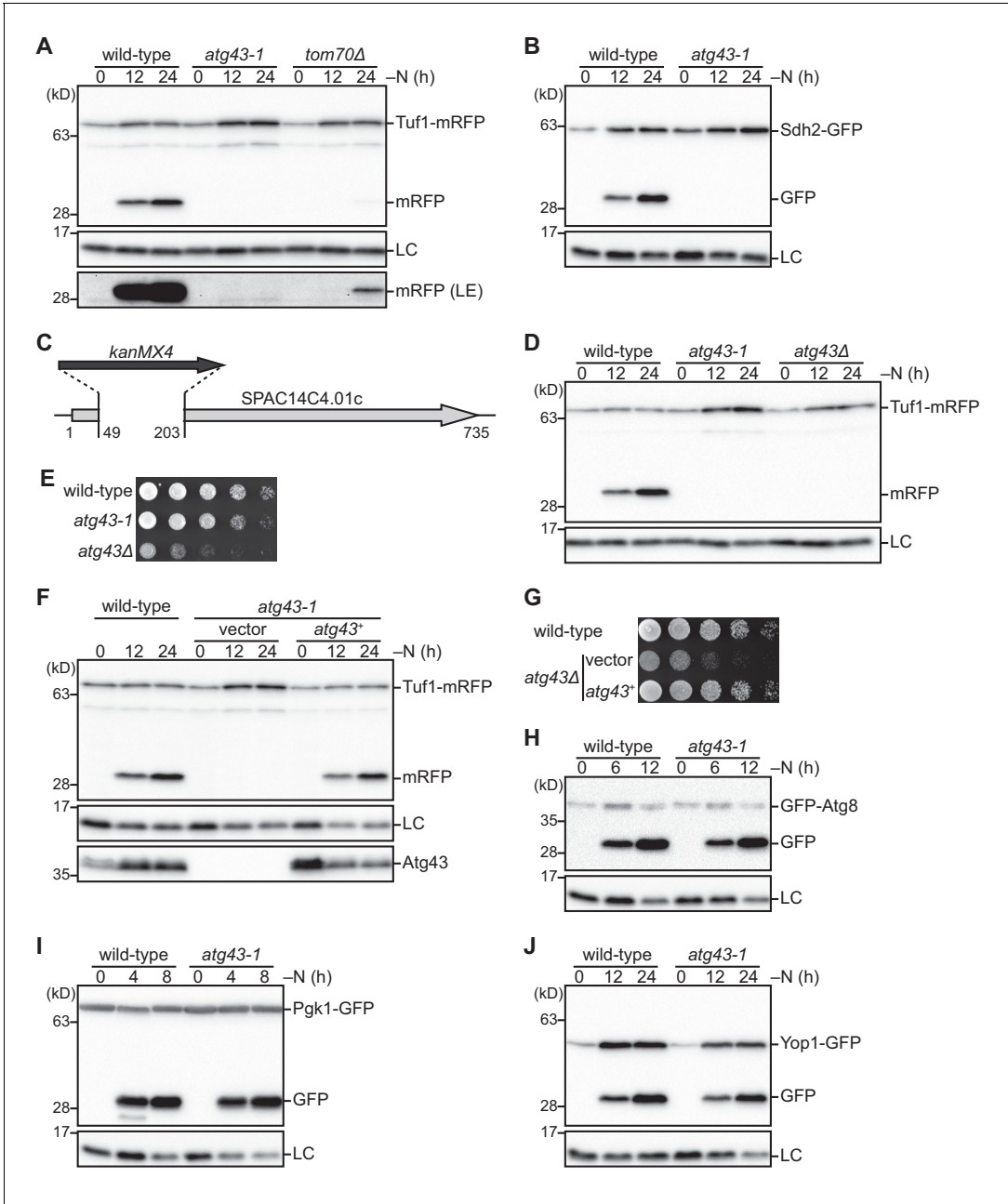

**Figure 1.** Atg43 is required for starvation-induced mitochondrial degradation in *S. pombe*. (**A, D**) The indicated strains expressing Tuf1-mRFP were grown in EMM and shifted to the same medium without a nitrogen source (EMM–N). Cells were collected at the indicated time points after nitrogen starvation was initiated. Tuf1-mRFP processing was monitored by immunoblotting. Lower panel, long exposure (LE). (**B**) The indicated strains expressing Sdh2-GFP were collected at the indicated time points after shifting to nitrogen starvation medium. The processing of Sdh2-GFP was monitored by immunoblotting. (**C**) Schematic representation of the *atg43-1* allele. The protein-coding region is depicted by a gray arrow. The nucleotide region that is deleted by the replacement with the *kanMX4* gene (black arrow) is shown. (**E, G**) The indicated strains were grown in EMM, and their serial dilutions were spotted onto solid YES medium for the growth assay. (**F**) The protein-coding region of the *atg43*⁺ gene was cloned into an integration vector to express *atg43*⁺ under the control of the moderate *adh31* promoter. The *atg43-1* mutants integrated with the *atg43*⁺ gene (*atg43*⁺) or an empty vector (vector), as well as a wild-type strain, were collected at the indicated time points after induction of nitrogen starvation and were subsequently subjected to immunoblotting. (**H–J**) The indicated strains expressing GFP-Atg8, Pgk1-GFP, or Yop1-GFP were collected at the indicated time points after induction of nitrogen starvation. Degradation of each substrate was monitored by immunoblotting. Histone H3 was used as a loading control (LC) for immunoblotting.

The online version of this article includes the following figure supplement(s) for figure 1:

**Figure supplement 1.** Mitochondrial proteins are degraded by autophagy dependently on Atg43 and Tom70.

strains expressing GFP-tagged Sdh2, Tom70, and Mic60, which are mitochondrial matrix, outer membrane, and inner membrane proteins, respectively. As in the case of Tuf1-mRFP, the processing of these proteins upon nitrogen starvation was impaired in the *atg43-1* mutant (*Figure 1B* and *Figure 1—figure supplement 1C and D*), indicating that the mutant is defective in degradation of various compartments of the mitochondria.

As a large part of the *atg43⁺* gene remains in the *atg43-1* allele (*Figure 1C*), we constructed mutants that lacked the whole protein-coding region of *atg43⁺* (*atg43Δ*) using gene targeting. We confirmed that the *atg43Δ* mutant is defective in Tuf1-mRFP processing upon nitrogen starvation (*Figure 1D*). Remarkably, *atg43Δ* cells exhibited a growth defect, whereas *atg43-1* cells grew nearly normally (*Figure 1E*). Since autophagy-defective mutants are proficient in cell growth (*Figure 1—figure supplement 1E*), likely, a process besides mitochondrial degradation is also defective in the *atg43Δ* mutant.

To examine whether the *atg43⁺* gene product (Atg43) is responsible for the observed phenotypes, we cloned the protein-coding region of the *atg43⁺* gene under a moderate constitutive promoter (*Padh31*) and integrated it at an ectopic genomic locus. Expression of Atg43 restored the processing of Tuf1-mRFP in *atg43-1* cells (*Figure 1F*). We used an antibody raised against the N-terminal 150 amino acid (aa) region of Atg43 and detected proteins of the same size in both wild-type cells and the *atg43-1* cells that had ectopic expression of Atg43 (*Figure 1F*). These findings suggest that Atg43 is responsible for mitochondrial degradation. We also confirmed that expression of Atg43 restores cell growth in the *atg43Δ* mutant (*Figure 1G*). Collectively, these data suggest that Atg43 has dual functions, one in the autophagic degradation of mitochondria and the other in a cellular process that affects cell growth. Hereafter, we used *atg43-1* as a separation-of-function mutant predominantly defective in mitochondrial degradation.

## Autophagic degradation of mitochondria occurs selectively in fission yeast

To determine whether Atg43 is required for autophagic degradation of substrates other than mitochondria, bulk autophagic activity was assessed by measurement of GFP-Atg8 processing (*Mukaiyama et al., 2010*). The *atg43-1* mutant produced free GFP (*Figure 1H*), suggesting that bulk autophagy is normal. This was confirmed by assessing the processing of GFP-tagged cytoplasmic proteins, Pgk1 (*Figure 1I*) and Tdh1 (*Figure 1—figure supplement 1F*). Next, to determine whether Atg43 is involved in the degradation of another organelle, we examined degradation of the endoplasmic reticulum (ER) by monitoring a GFP-tagged ER membrane protein, Yop1 (*Zhang et al., 2010*). In the *atg43-1* mutant, Yop1-GFP processing was detected (*Figure 1J*). These findings suggest that Atg43 is specific to mitochondrial degradation, indicating that mitochondria are selectively degraded Atg43-dependently. We also confirmed that bulk and ER autophagy occur in the absence of Tom70 (*Figure 1—figure supplement 1G–I*), indicating that Tom70 is also a mitochondria-specific factor. Thus, we hereafter refer to the selective autophagic degradation of mitochondria in fission yeast as mitophagy.

## Atg43 is a transmembrane protein anchored to the MOM

Atg43 is conserved among *Schizosaccharomyces* species (*Figure 2—figure supplement 1A*), and Atg43-like sequences are present in fungi, but the homology is limited to the C-terminal region (*Figure 2—figure supplement 1B*). We could not identify proteins orthologous to Atg43 in *S. cerevisiae* or mammals.

Immunoblotting of Atg43 in *S. pombe* detected a signal larger than the predicted size of 27 kDa (*Figure 1F*). Upon nitrogen starvation, Atg43 protein levels were increased (*Figure 2A and B*). Under nutrient-rich conditions, target of rapamycin complex 1 (TORC1) suppresses autophagy through phosphorylation of the core autophagy proteins (*Noda, 2017*). We inactivated TORC1 using a temperature-sensitive mutant of the Tor2 kinase subunit of TORC1 (*tor2-13*) (*Uritani et al., 2006*). At the non-permissive temperature for *tor2-13*, Atg43 levels increased (*Figure 2C*) and mitophagy took place (*Figure 2—figure supplement 2A*), suggesting that Atg43 expression and mitophagy are regulated by TORC1 signaling. In cells constitutively expressing Atg43 under the *adh31* (*Figure 1F*) or *nmt1* (*Figure 2—figure supplement 2B*) promoter, nitrogen starvation was required for mitophagy induction, indicating that Atg43

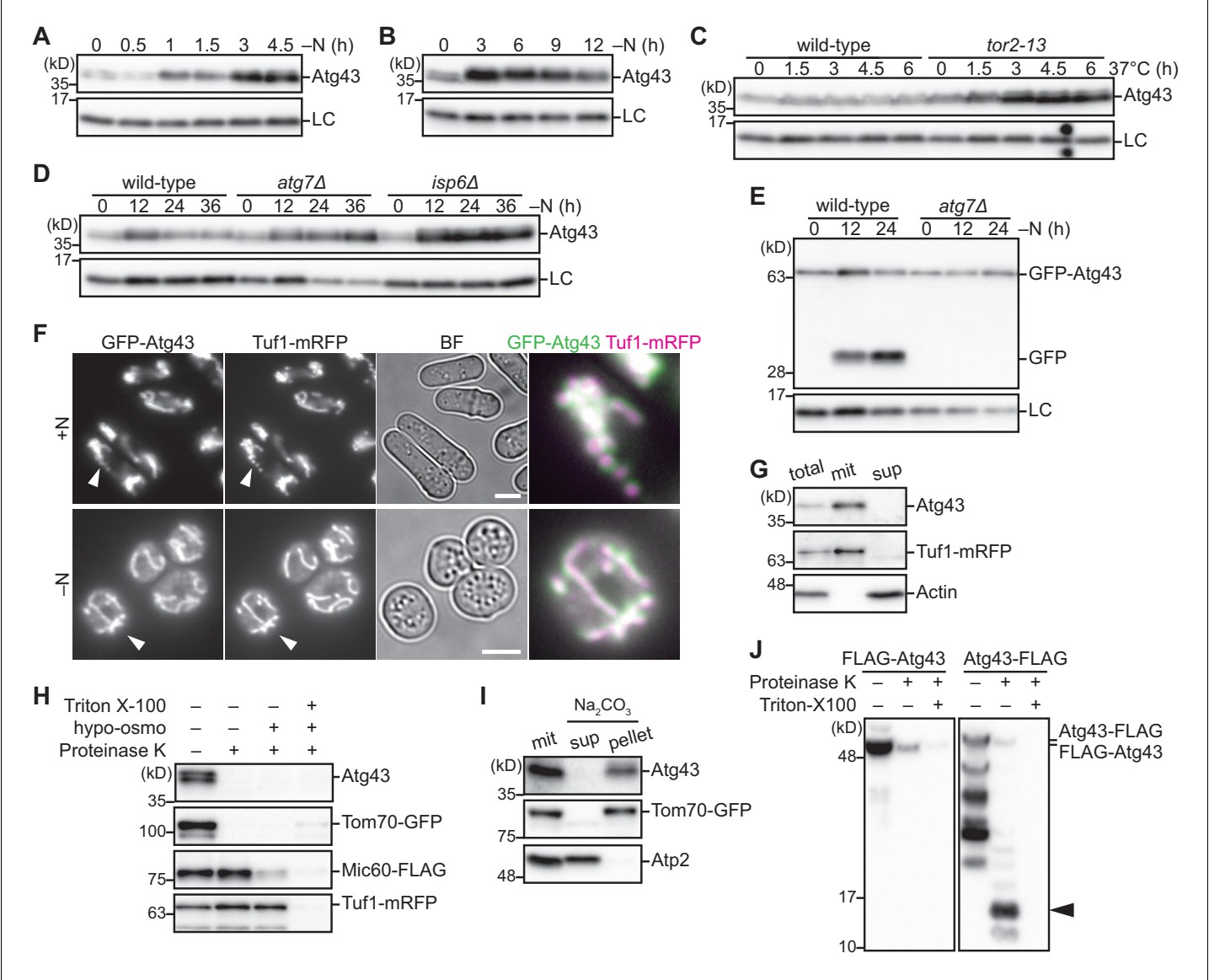

**Figure 2.** Atg43 is a transmembrane protein localized to the MOM. (**A, B**) Wild-type cells were grown in EMM and shifted to EMM–N for nitrogen starvation. Cells were collected at the indicated time points after shifting to nitrogen starvation medium and were then subjected to immunoblotting. (**C**) The indicated strains were grown in EMM at 26°C. The temperature was then shifted to 37°C to inactivate TORC1. Cells were collected at the indicated time points after the temperature shift and were used for immunoblotting. (**D, E**) The indicated strains were collected at the indicated time points after shifting to nitrogen starvation medium and were used for immunoblotting. (**F**) Cells expressing GFP-Atg43 and Tuf1-mRFP were grown in EMM (+N) or cultured in EMM–N for 12 hr (–N) for microscopy. A magnified view of the indicated area is shown in the right panel. Scale bars represent 5 μm. BF, bright-field image. (**G**) Fractionation was conducted using cells expressing Tuf1-mRFP. The total cell homogenate (total) was fractionated by centrifugation to obtain a mitochondria-enriched pellet (mit) and supernatant (sup). Tuf1-mRFP and actin were detected as markers of the mitochondria and cytosol, respectively. (**H**) A mitochondrial fraction was prepared from cells expressing Tom70-GFP, Mic60-FLAG, and Tuf1-mRFP, treated with (+) or without (–) proteinase K, under different conditions. Hypo-osmotic swelling resulted in rupture of the MOM and the detergent Triton X-100 lysed mitochondria. Tom70-GFP, Mic60-FLAG, and Tuf1-mRFP were detected as markers of the mitochondrial outer membrane, inner membrane, and matrix, respectively. (**I**) The mitochondrial fraction (mit) was prepared from cells expressing Tom70-GFP, treated with sodium carbonate, and separated into the soluble supernatant (sup) and membrane pellet (pellet) by centrifugation. Tom70-GFP and Atp2 were detected as markers for integral and peripheral membrane proteins, respectively. (**J**) The mitochondrial fraction was prepared from cells expressing N-terminal or C-terminal FLAG-tagged Atg43 (FLAG-Atg43 or Atg43-FLAG, respectively), followed by treatment with (+) or without (–) proteinase K. The Triton X-100 detergent lysed mitochondria. The proteinase-resistant band is indicated by the arrowhead. Histone H3 was used as a loading control (LC) for immunoblotting.

The online version of this article includes the following figure supplement(s) for figure 2:

**Figure supplement 1.** The C-terminal region of Atg43 is conserved.

**Figure supplement 2.** Atg43 is a transmembrane protein that is expressed during nitrogen starvation.

expression is not sufficient to induce mitophagy. In cells defective in autophagy, Atg43 accumulated during nitrogen starvation (*Figure 2D*), suggesting that Atg43 is efficiently degraded by autophagy. We further confirmed autophagy-dependent degradation of Atg43 using a GFP processing assay (*Figure 2E*). Thus, we identified that Atg43 is upregulated during the initiation of autophagy and is degraded as autophagy proceeds.

To determine the subcellular localization of Atg43, we assessed cells expressing GFP-tagged Atg43. Atg43 was detected on mitochondria that were labeled with Tuf1-mRFP under both nitrogen-rich and starvation conditions (*Figure 2F*). Mitochondrial localization of Atg43 was further confirmed after fractionation of cell homogenates, where Atg43 was detected in the mitochondria-enriched fraction, along with Tuf1-mRFP (*Figure 2G*). We subsequently treated the mitochondria-enriched fraction with proteinase K. In contrast to Mic60 and Tuf1, markers of the mitochondrial inner membrane and matrix, respectively, Atg43 was digested without hypo-osmotic swelling or Triton X-100 lysis, similarly to Tom70 (*Figure 2H*). This suggests that Atg43 is localized to the MOM. Using the HMMTOP prediction method (*Tusnády and Simon, 2001*), we identified that Atg43 is predicted to contain a transmembrane domain at its C-terminus (*Figure 2—figure supplement 1B*). After centrifugation of the alkaline-treated mitochondrial fraction, Atg43 was detected in the pellet along with Tom70, an integral MOM protein, whereas the peripheral inner membrane protein, Atp2, was detected in the supernatant (*Figure 2I*). Therefore, Atg43 is likely to be an integral membrane protein. To determine the membrane topology, the mitochondria-enriched fraction prepared from cells expressing N-terminal or C-terminal FLAG-tagged Atg43 was treated with proteinase K. Anti-FLAG immunoblotting detected a detergent-sensitive signal of 10 to 17 kDa for C-terminal-tagged but not for N-terminal-tagged Atg43 (*Figure 2J*). A similar result was obtained in cells expressing the C-terminal 80 aa region of Atg43 that contains the predicted transmembrane domain (*Figure 2—figure supplement 2C and D*). These data support the hypothesis that the C-terminus of Atg43 is protected inside mitochondria. Thus, we propose that Atg43 is a transmembrane protein localized to the MOM with its N-terminus facing the cytosol.

## Atg43 harbors an AIM in its cytosolic region that is essential for mitophagy

To identify the functional regions in Atg43, we prepared cells that expressed an N-terminal truncated form of Atg43 (*Figure 3A*) and tested their ability to perform mitophagy. Although the 20 aa N-terminal region was dispensable for mitophagy, mitophagy was completely abolished in cells expressing Atg43 lacking the 40 aa N-terminal region (*Figure 3B*), suggesting that aa 21–40 of Atg43 contain a factor required for mitophagy. This region harbors $Y^{28}ELI^{31}$, an AIM/LIR (W/F/Y-x-x-L/I/V) sequence, which is conserved among selective autophagy receptor/adaptor proteins. To determine the role of the AIM in mitophagy, we assessed mitophagy in cells expressing AIM-mutated Atg43 ($A^{28}ELA^{31}$, $\Delta AIM$) and found that these cells were defective in mitophagy (*Figure 3C*). Next, we investigated the interaction between the cytosolic region of Atg43 and Atg8 using a yeast two-hybrid assay. The 184 aa at the N-terminal of Atg43 and aa 1–40 of Atg43 resulted in a positive signal with Atg8 (*Figure 3D*). By contrast, the AIM-mutated aa 1–40 and 1–184 of Atg43 did not exhibit a positive signal (*Figure 3E*). Furthermore, interaction between Atg8 and Atg43 was confirmed by immunoprecipitation (*Figure 3F*), and the detected interaction was dependent on the AIM (*Figure 3G*). Taken together, these data indicate that Atg43 can interact with Atg8 AIM-dependently. Interaction of Atg43 with another core autophagy protein, Atg11 was also examined by immunoprecipitation, and unlike Atg8, Atg11 did not efficiently co-precipitate Atg43 (*Figure 3—figure supplement 1A*).

We also monitored the mitophagy ability of strains expressing Atg43 that lacks the central region (*Figure 3A*). Mitophagy still occurred in cells expressing Atg43 with deletion in aa 41–80, 81–120, and 121–164 (*Figure 3H*), suggesting that aa 41–164 of Atg43 are not essential for mitophagy.

Besides mitophagy, cell growth was assessed with the N-terminal truncated Atg43 mutants. Cells expressing Atg43 that lacks the 164 N-terminal aa grew normally (*Figure 3I*). By contrast, when the N-terminal truncation reached the conserved region (aa 165–184), cell growth was impaired (*Figure 3I*), indicating that aa 165–184 of Atg43 contain a factor that is necessary for normal cell growth. We also assessed mitochondrial localization and found that with deletion of the N-terminal 164 and 184 aa, mitochondrial localization was retained (*Figure 3J and K*). Taken together, these findings suggest that aa 21–40 of Atg43, which contain an AIM, are necessary for mitophagy, but

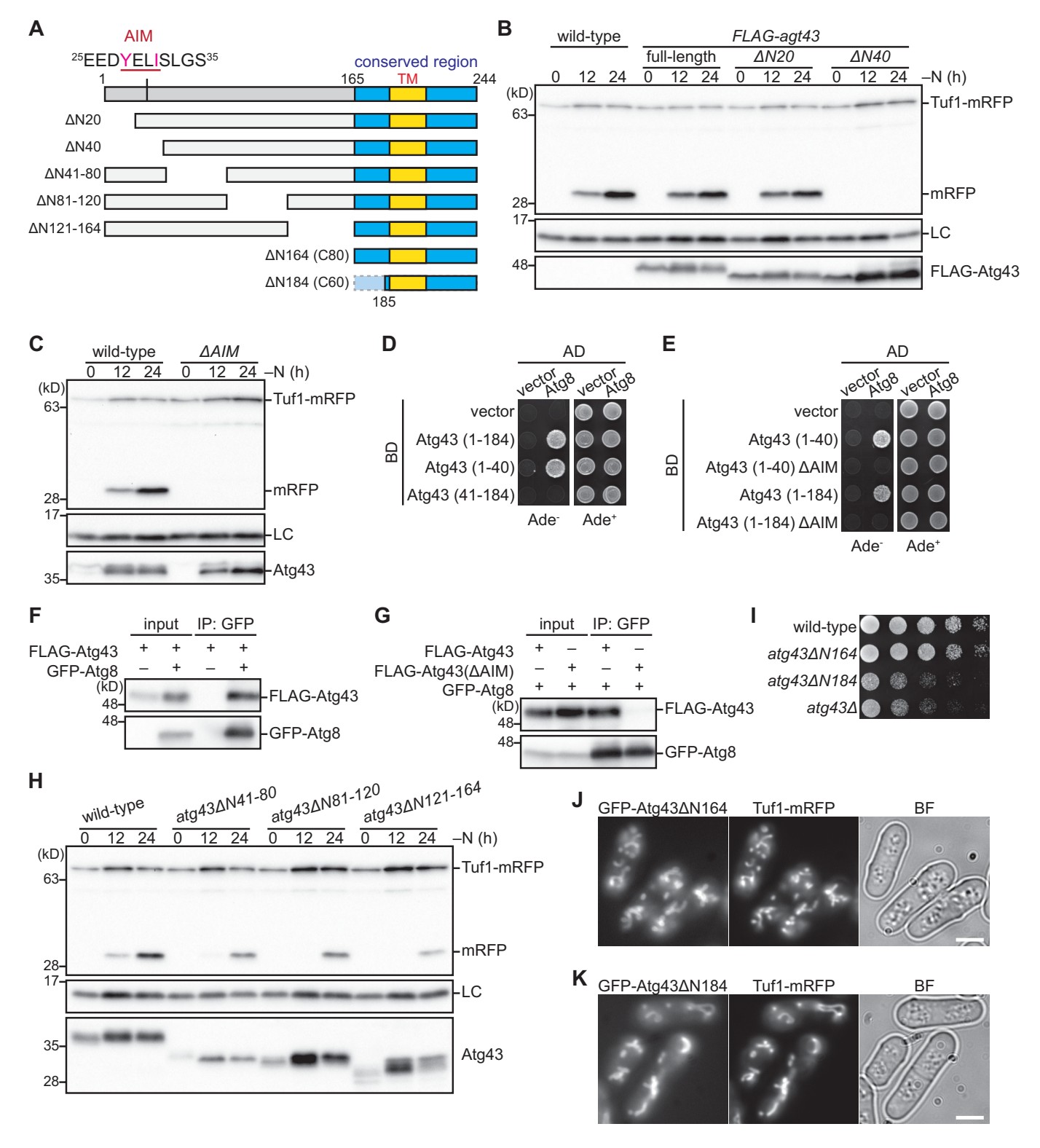

**Figure 3.** Atg43 contains an AIM that is essential for mitophagy. (**A**) Schematic representation of N-terminal deleted forms of Atg43. The region conserved among fungi is highlighted in blue. The N-terminal AIM and C-terminal predicted transmembrane domain (TM) are indicated. (**B**) Wild-type cells and strains expressing FLAG-tagged Atg43, with or without N-terminal truncation, under the thiamine-repressible *nmt1* promoter, were grown in EMM containing thiamine for moderate expression. Cells were collected at the indicated time points after shifting to nitrogen starvation medium and

*Figure 3 continued on next page*

*Figure 3 continued*

were subsequently used for immunoblotting. (C) Cells expressing wild-type and AIM-mutated (ΔAIM) Atg43 from the native gene locus were collected at the indicated time points after shifting to nitrogen starvation medium and subjected to immunoblotting. (D, E) A yeast two-hybrid assay between activation domain (AD)-fused Atg8 and DNA-binding domain (BD)-fused Atg43 fragments, without or with mutations in the N-terminal AIM (ΔAIM). (F, G) Crude cell lysates (input) were prepared from cells with (+) or without (–) expression of GFP-fused Atg8 and FLAG-tagged Atg43 or AIM-mutated Atg43 (ΔAIM). Anti-GFP immunoprecipitants (IP: GFP) were analyzed by immunoblotting. Cells were collected 12 hr after shifting to nitrogen starvation medium and were subjected to immunoprecipitation. (H) Cells expressing wild-type and deleted forms of Atg43 from the native gene locus were collected at the indicated time points after shifting to nitrogen starvation medium and were subsequently used for immunoblotting. (I) The indicated strains were grown in EMM, and their serial dilutions were spotted onto solid YES medium for the growth assay. (J, K) Cells expressing N-terminal truncated forms of Atg43 tagged with GFP were grown in EMM for microscopy. Tuf1-mRFP was detected as a mitochondrial marker. Scale bars represent 5 μm. BF, bright-field image. Histone H3 was used as a loading control (LC) for immunoblotting.

The online version of this article includes the following figure supplement(s) for figure 3:

**Figure supplement 1.** The N-terminal region of Atg43 interacts with Atg8 and is dispensable for the mitochondrial localization.

dispensable for normal cell growth and mitochondrial localization of Atg43. These data are consistent with that the *atg43-1* mutation predominantly affects mitophagy. The *atg43-1* allele lacks the part coding for the AIM-containing region but retains the part coding for the C-terminal region of Atg43 (*Figure 1C*). Indeed, the *atg43-1* mutant expressed a truncated form of Atg43 that was enriched in mitochondria (*Figure 3—figure supplement 1B and C*).

## The primary role of Atg43 in mitophagy is to anchor the AIM to mitochondria

To investigate the role of the conserved C-terminal region of Atg43, cells expressing Atg43 lacking the C-terminal region were assessed (*Figure 4A*). Atg43 lacking the 20 C-terminal aa exhibited only a partial defect in mitophagy, whereas Atg43 with a truncation of the 60 C-terminal aa was defective in mitophagy (*Figure 4B*). The 60 C-terminal aa were also necessary for normal cell growth (*Figure 4C*). We also examined the subcellular localization of the truncation mutants. Atg43 lacking the 20 C-terminal aa remained localized to mitochondria, whereas Atg43 lacking the 60 C-terminal aa had diffuse localization in the cytoplasm (*Figure 4D*). Collectively, these results suggest that aa 185–224 of Atg43, which contain the predicted transmembrane domain, are necessary for mitophagy, normal cell growth, and mitochondrial localization of Atg43.

As shown above, the N-terminal portion of the conserved region (aa 165–184 from the N-terminus or aa 61–80 from the C-terminus) of Atg43 is required for normal cell growth (*Figure 3I*) but not for mitochondrial localization (*Figure 3K*). We further confirmed this using a mutant lacking this region (*atg43ΔN165-184*) (*Figure 4C and D*) and found that mitophagy occurred in Atg43ΔN165-184 expressing cells (*Figure 4B*). Thus, aa 165–184 of Atg43 are important for normal cell growth but is not essential for mitophagy.

Next, we aimed to determine whether the 60 aa C-terminal region of Atg43 has an additional role in mitophagy besides mitochondrial localization. To this end, we used the MOM protein, Fis1, and fused its N-terminus to a GFP-binding protein (GBP) (*Chen et al., 2017*). When Atg43 lacking the 60 C-terminal aa was fused to GFP (Atg43ΔC60-GFP) and expressed in control cells (*fis1*[+]), the GFP signal was observed in the cytoplasm (*Figure 4E*). By contrast, when Atg43ΔC60-GFP was co-expressed with GBP-fused Fis1 (*GBP-fis1*), Atg43ΔC60-GFP was detected on the mitochondria, due to the interaction between GFP and GBP (*Figure 4E*). Remarkably, under these conditions, mitophagy occurred upon nitrogen starvation (*Figure 4F*, *GBP-fis1 atg43ΔC60-GFP*). We further confirmed that mitophagy is absent in cells that solely express Atg43ΔC60-GFP or GBP-Fis1 (*Figure 4F*). Therefore, we identified that a major function of the 60 aa C-terminal region of Atg43 is to anchor Atg43 to the MOM for mitophagy and found that this activity can be replaced by another MOM-anchoring protein.

We have thus far identified that aa 21–40 and 185–224 of Atg43 are necessary for mitophagy and mitochondrial localization, respectively. Next, we investigated whether these regions are sufficient for mitophagy and mitochondrial localization, respectively. Microscopy of GFP-tagged aa 185–224 of Atg43 revealed that this region is sufficient for mitochondrial localization (*Figure 4G*). Artificial tethering of aa 21–40 of Atg43 to mitochondria using the interaction between GFP and GBP-fused Fis1 restored mitophagy in Atg43-deficient cells (*Figure 4H*), suggesting that mitochondrial

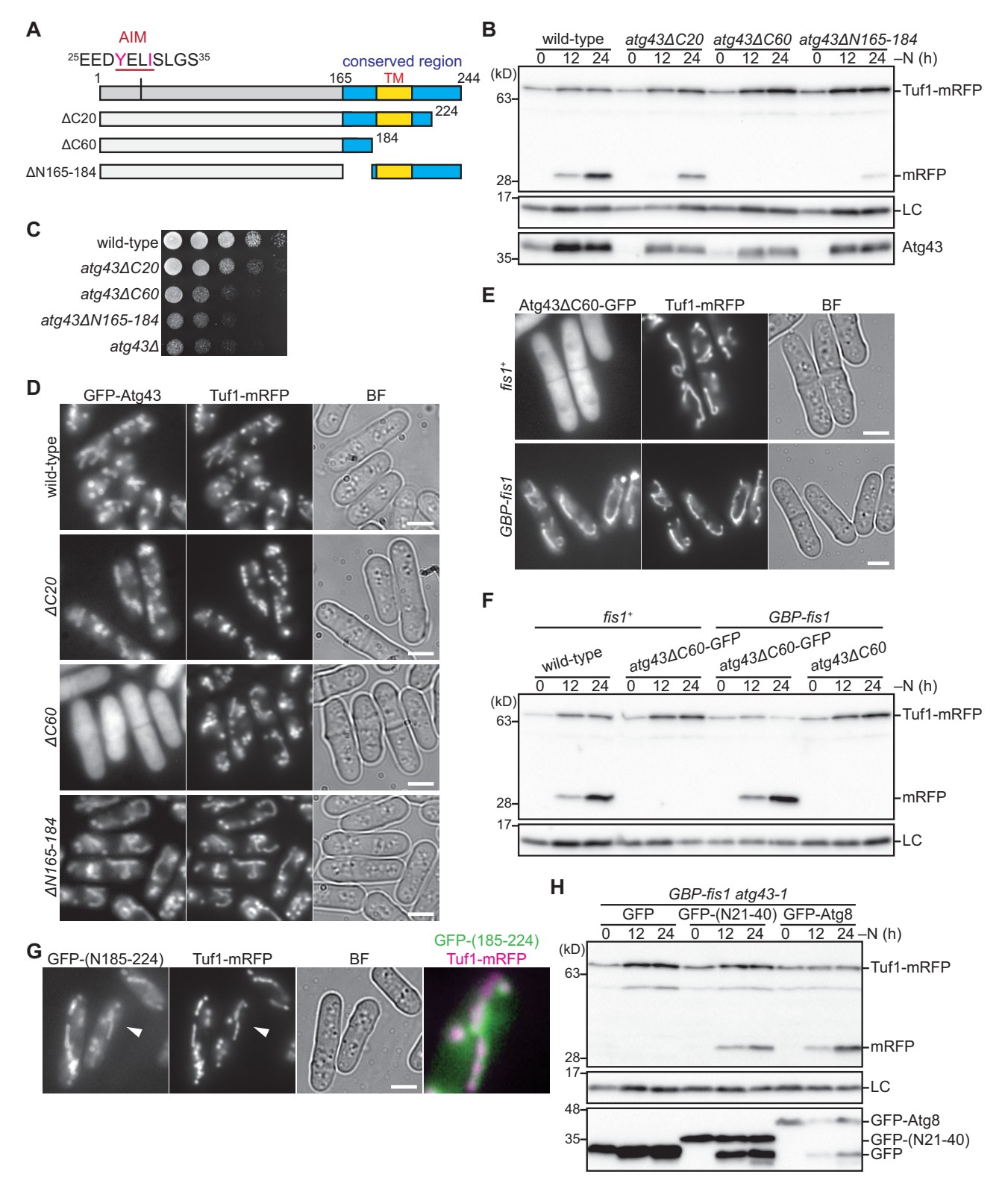

**Figure 4.** The C-terminal region of Atg43 that contains a predicted transmembrane domain is required for mitochondrial localization. (**A**) Schematic representation of C-terminal deleted forms of Atg43. The region conserved among fungi is highlighted in blue. The N-terminal AIM and predicted C-terminal transmembrane domain (TM) are indicated. (**B**) Cells expressing wild-type and C-terminal deleted forms of Atg43 from the native gene locus were collected at the indicated time points after shifting to nitrogen starvation medium and were subjected to immunoblotting. Each strain contained

*Figure 4 continued on next page*

*Figure 4 continued*

an integration vector expressing the mitophagy-defective form of Atg43, that lacks the 164 N-terminal aa, to maintain a normal growth rate. (C) The indicated strains were grown in EMM, and their serial dilutions were spotted onto solid YES medium for a growth assay. (D) Cells expressing wild-type and C-terminal truncated forms of GFP-tagged Atg43 were grown in EMM for microscopy. Tuf1-mRFP was detected as a mitochondrial marker. (E) Wild-type (*fis1*⁺) and *GBP-fis1* cells co-expressing GFP-tagged Atg43 with a 60 aa deletion in the C-terminus and Tuf1-mRFP were grown in EMM for microscopy. (F) Wild-type (*fis1*⁺) and *GBP-fis1* cells expressing full-length (wild-type) or C-terminal truncated Atg43 (Atg40ΔC60) with or without GFP fusion were collected at the indicated time points after shifting to nitrogen starvation medium. Cells were then used for immunoblotting. (G) Cells expressing the GFP-fused aa 185–224 of Atg43 and Tuf1-mRFP were grown in EMM for microscopy. A magnified view of the indicated area is shown in the right panel. (H) Mitophagy-defective *atg43-1* cells with GBP-fused Fis1 were transformed to express GFP, GFP-fused aa 21–40 of Atg43, or GFP-fused Atg8. The indicated strains were collected at the indicated time points after shifting to nitrogen starvation medium and used for immunoblotting. Histone H3 was used as a loading control (LC) for immunoblotting. Scale bars represent 5 μm. BF, bright-field image.

anchoring of the 20 aa region containing AIM is sufficient for mitophagy. Notably, artificial loading of Atg8 to mitochondria also rescued mitophagy (*Figure 4H*). Therefore, we propose that a major role of Atg43 in the mitophagy process is to tether Atg8 to mitochondria through direct interaction with Atg8 via the AIM region.

## The MIM complex facilitates mitophagy through the loading of Atg43 to the MOM

To identify proteins that physically interact with the conserved C-terminal region of Atg43, cells expressing a FLAG-tagged version of the 80 C-terminal aa of Atg43 (aa 165–244) were subjected to anti-FLAG affinity purification. The resulting immunoprecipitants were analyzed using mass spectrometry and the Mim2 protein was identified (*Figure 5—figure supplement 1A*). The interaction between full-length Atg43 and Mim2 was confirmed using reciprocal immunoprecipitation experiments (*Figure 5A* and *Figure 5—figure supplement 1B*). In budding yeast, Mim2 is a component of the mitochondrial import machinery (MIM) complex, which consists of Mim1 and Mim2 (*Dimmer et al., 2012*; *Stefan Dimmer and Rapaport, 2010*). Reciprocal immunoprecipitation experiments revealed that Mim2 also interacts with Mim1 in fission yeast (*Figure 5—figure supplement 1C and D*), suggesting that the MIM complex is conserved. We further confirmed the interaction between Atg43 and Mim1 using reciprocal immunoprecipitation experiments (*Figure 5B* and *Figure 5—figure supplement 1E*). Taken together, these data indicate that Atg43 interacts with the MIM complex.

To examine whether the MIM complex is involved in the function of Atg43, we analyzed the *mim1Δ* and *mim2Δ* mutants. The *mim1Δ* and *mim2Δ* mutants exhibited a sever growth defect (*Figure 5—figure supplement 1F*). As Mim1 and Mim2 are thought to be involved in the insertion of transmembrane proteins into the MOM, we then assessed the mitochondrial localization of Atg43. In the absence of Mim1 or Mim2, the GFP-Atg43 signal at the mitochondria was severely decreased (*Figure 5C*). We also found that Atg43 was undetectable by immunoblotting in the *mim1Δ* and *mim2Δ* mutants (*Figure 5—figure supplement 1G*), indicating that Atg43 becomes unstable in these mutants. These findings suggest that stable localization of Atg43 to the MOM is highly dependent on the MIM complex. This raises the possibility that the MIM complex assists Atg43 through facilitating its mitochondrial localization. Consistent with this, mitophagy was impaired in the *mim1Δ* and *mim2Δ* mutants (*Figure 5D*).

As Tom70, another mitochondrial import factor, was also identified as a protein required for mitophagy (*Figure 1A*), we assessed the relationship between Tom70, Atg43, and the MIM complex. In budding yeast, Tom70 cooperates with the MIM complex to sort MOM proteins. The mitochondrial loading of Tom70 is also facilitated by the MIM complex (*Becker et al., 2008*). We confirmed that Mim1 and Mim2 are required for stable localization of Tom70 on mitochondria in fission yeast (*Figure 5—figure supplement 1H*). We then examined the interdependency between Atg43 and Tom70 for mitochondrial localization. Atg43 was observed on mitochondria in the absence of Tom70 (*Figure 5E*) and vice versa (*Figure 5—figure supplement 1I*), suggesting that their mitochondrial import is mutually independent. We also found that neither Atg43 nor Tom70 was required for the localization of the MIM complex to mitochondria (*Figure 5—figure supplement 1J and K*).

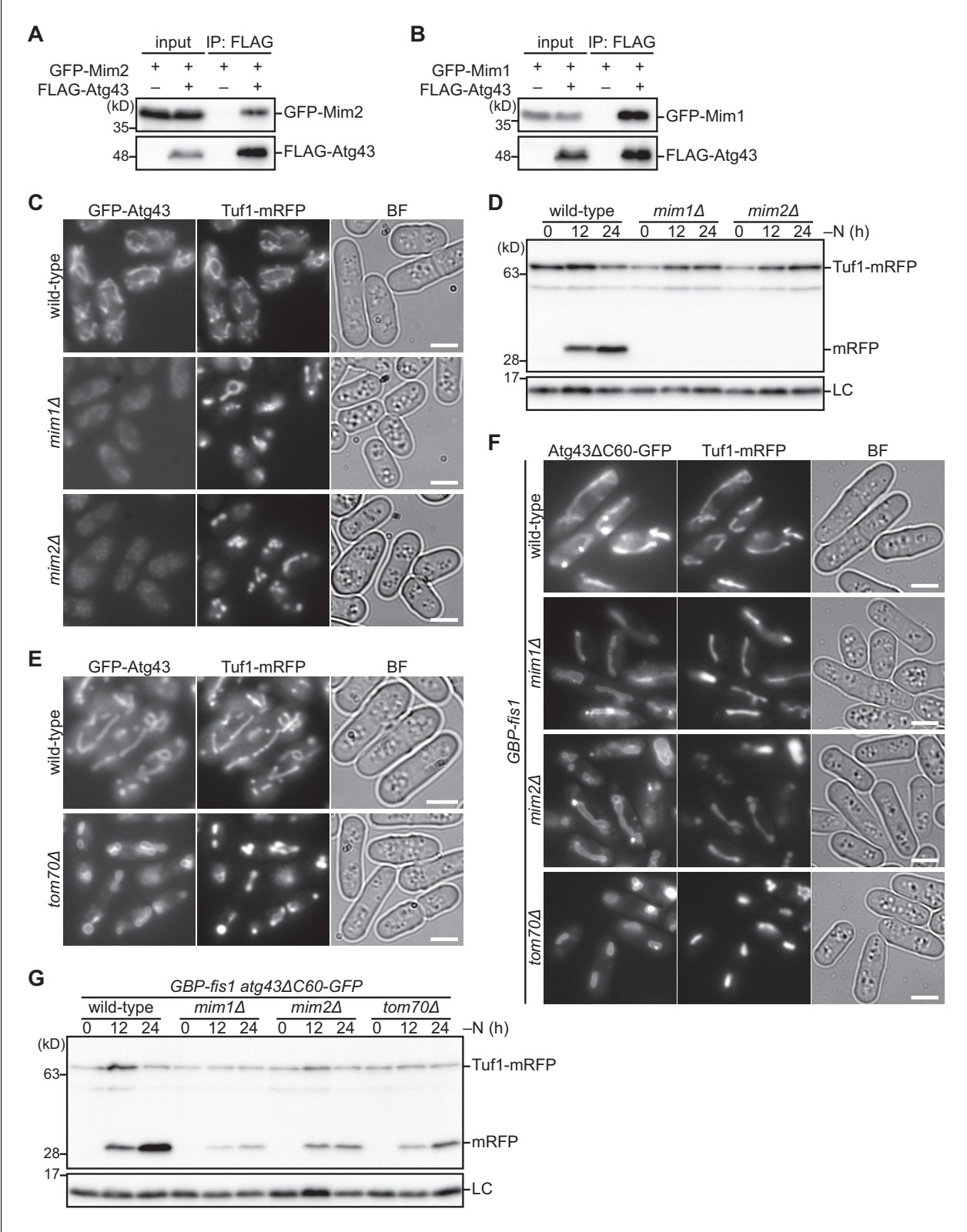

**Figure 5.** The MIM complex facilitates mitophagy through the loading of Atg43 to the MOM. (**A, B**) Crude cell lysates (input) were prepared from cells co-expressing GFP-fused Mim2 or Mim1 with (+) or without (–) FLAG-tagged Atg43. Anti-FLAG immunoprecipitants (IP: FLAG) were analyzed by immunoblotting. (**C, E**) The indicated strains co-expressing GFP-Atg43 and Tuf1-mRFP were grown in EMM for microscopy. (**D**) The indicated strains expressing Tuf1-mRFP were collected at the indicated time points after shifting to nitrogen starvation medium and were used for immunoblotting. (**F**)
*Figure 5 continued on next page*

Figure 5 continued

The indicated strains co-expressing a GFP-fused C-terminal truncated form of Atg43 (Atg43ΔC60-GFP) and Tuf1-mRFP were grown in EMM for microscopy. (G) Wild-type, *mim1Δ*, *mim2Δ*, and *tom70Δ* cells co-expressing GBP-fused Fis1 and GFP-fused Atg43 lacking the 60 C-terminal aa (Atg43ΔC60-GFP) were collected at the indicated time points after shifting to nitrogen starvation medium. Cells were then used for immunoblotting. Histone H3 was used as a loading control (LC) for immunoblotting. Scale bars represent 5 μm. BF, bright-field image.

The online version of this article includes the following figure supplement(s) for figure 5:

**Figure supplement 1.** The MIM complex facilitates the mitochondrial localization of Atg43 and Tom70.

We thus hypothesized that the primary role of the MIM complex in mitophagy is to facilitate the integration of Atg43 to the MOM. To test this hypothesis, we made use of the artificial tethering of GFP-tagged Atg43 to mitochondria by GBP-fused Fis1. In budding yeast, mitochondrial integration of Fis1 is independent of the MIM complex and Tom70 (*Dimmer et al., 2012*; *Kemper et al., 2008*). As expected, when Atg43ΔC60-GFP was co-expressed with GBP-Fis1, it was observed on mitochondria in the absence of Mim1, Mim2, or Tom70 (*Figure 5F*) and mitophagy took place upon nitrogen starvation (*Figure 5G*). This indicates that artificial tethering of Atg43 to mitochondria can override the requirement of the mitochondrial import factors for mitophagy. Thus, the MIM complex is likely to promote mitophagy through facilitation of the MOM integration of Atg43. Moreover, Tom70 may contribute to mitophagy possibly through the biogenesis of functional Atg43 on the MOM (see Discussion).

## Interaction between Atg43 and the MIM complex is important for normal cell growth but dispensable for mitophagy

Although the MIM complex contributes to the mitochondrial import of both Atg43 and Tom70, its interaction with Atg43 appeared much stronger than that with Tom70. First, Mim2 was efficiently co-immunoprecipitated with Atg43 and Mim1 but not with Tom70 (*Figure 6—figure supplement 1A*). Second, Atg43 was efficiently co-immunoprecipitated with Mim1 and Mim2 but not with Tom70 (*Figure 6—figure supplement 1B*). Third, in the absence of the MIM complex, the Tom70 protein was partially reduced, whereas Atg43 was undetectable (*Figure 5—figure supplement 1G*). These findings suggest that Atg43 forms a complex with the Mim proteins.

We constructed Atg43 deletion mutants to identify the region required for binding to the MIM complex (*Figure 6A*). While full-length and the 80 C-terminal aa (aa 165–244) of Atg43 interacted with Mim2, deletion of aa 165–184 impaired this interaction (*Figure 6B*). These findings indicate that aa 165–184 of Atg43 are important for its binding to the MIM complex. Truncation of the 60 C-terminal aa of Atg43, which contain the predicted transmembrane domain, also resulted in the loss of interaction with Mim2 (*Figure 6C and D*), suggesting that the mitochondrial localization of Atg43 is a prerequisite for this interaction.

Although aa 165–184 of Atg43 are required for the interaction with the MIM complex (*Figure 6B and D*), this region is not essential for mitochondrial localization of Atg43 (*Figures 3K*, *4D* and *6E*) or mitophagy (*Figures 4B* and *6E*). However, aa 165–184 of Atg43 are important for normal cell growth (*Figures 3I*, *4C* and *6E*). Taken together, these results suggest that the interaction between Atg43 and the MIM complex takes place after the MIM complex-mediated mitochondrial loading of Atg43 (*Figure 6F*). We propose that this interaction is important for the mitophagy-independent role of Atg43.

## Atg43-deficient cells can be used as a model of mitophagy-defective cells

Since *atg43-1* and AIM-mutated *atg43* cells predominantly exhibit a defect in mitophagy, we used the *atg43-1* and *atg43ΔAIM* mutants as models of mitophagy deficiency to examine the physiological roles of mitophagy. The *atg7Δ* mutant was also used as a model of autophagy deficiency. We first examined cell viability during nitrogen starvation with or without mitophagy/autophagy. Although the *atg43-1*, *atg43ΔAIM*, and *atg7Δ* mutants were viable on nutrient-rich medium 5 days after induction of nitrogen starvation, they exhibited reduced viability at 18 days (*Figure 7A*). This suggests that mitophagy is important for maintaining growth ability during long-term nitrogen starvation. Notably, *atg43-1* and *atg43ΔAIM* cells exhibited a severer growth defect than the *atg7Δ* mutant at

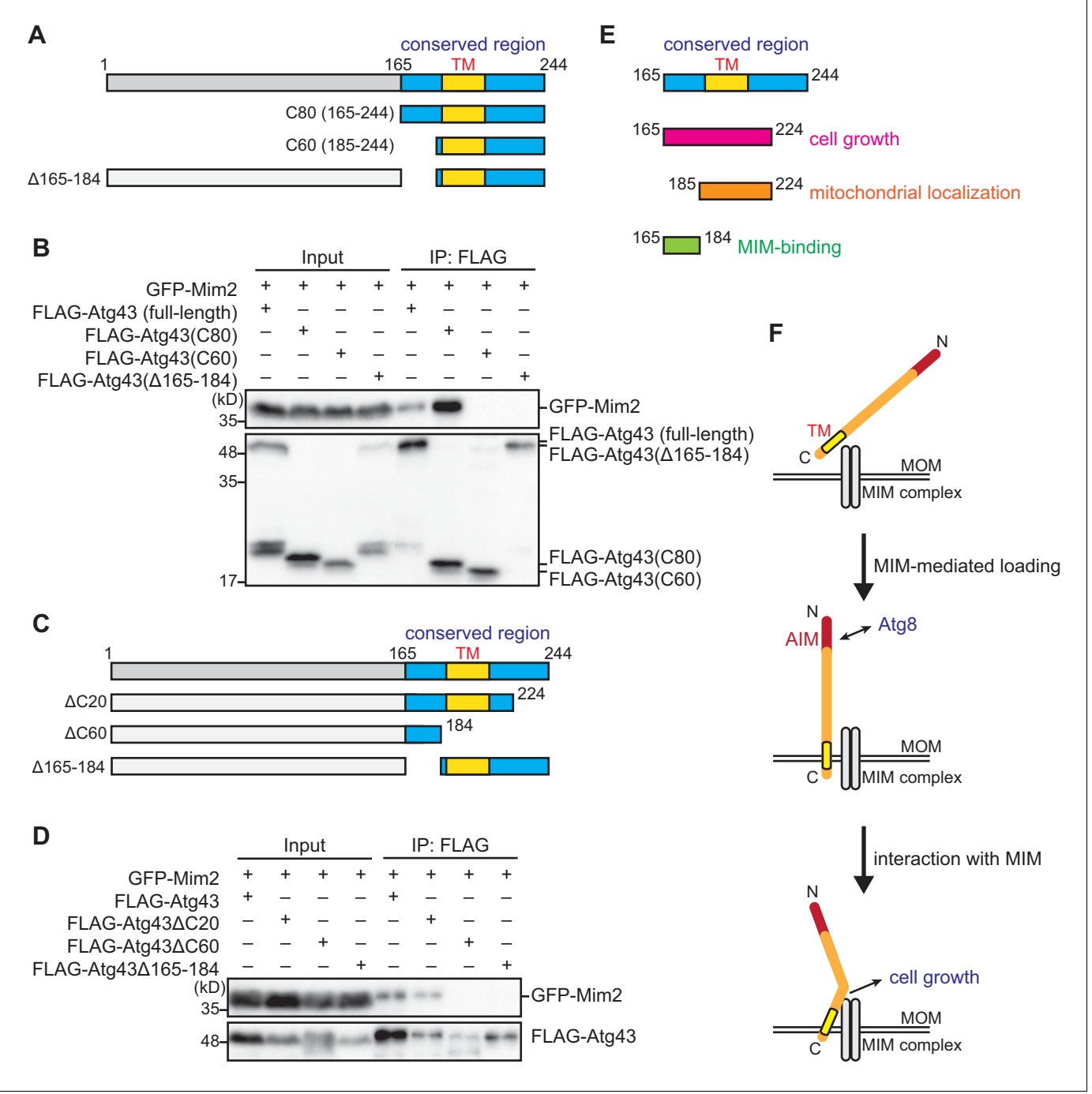

**Figure 6.** C-terminal region of Atg43 is required for its interaction with the MIM complex. (**A, C**) Schematic representation of deleted forms of Atg43. The region conserved among fungi is highlighted in blue. The C-terminal predicted transmembrane domain (TM) is also indicated. (**B, D**) Crude cell lysates (input) were prepared from cells co-expressing GFP-fused Mim2 and FLAG-tagged Atg43 with or without deletion. Anti-FLAG immunoprecipitants (IP: FLAG) were analyzed by immunoblotting. (**E**) Schematic representation of the C-terminal region of Atg43. (**F**) Schematic model describing the loading and function of Atg43 on the MOM.

The online version of this article includes the following figure supplement(s) for figure 6:

**Figure supplement 1.** Atg43 interacts with the MIM complex.

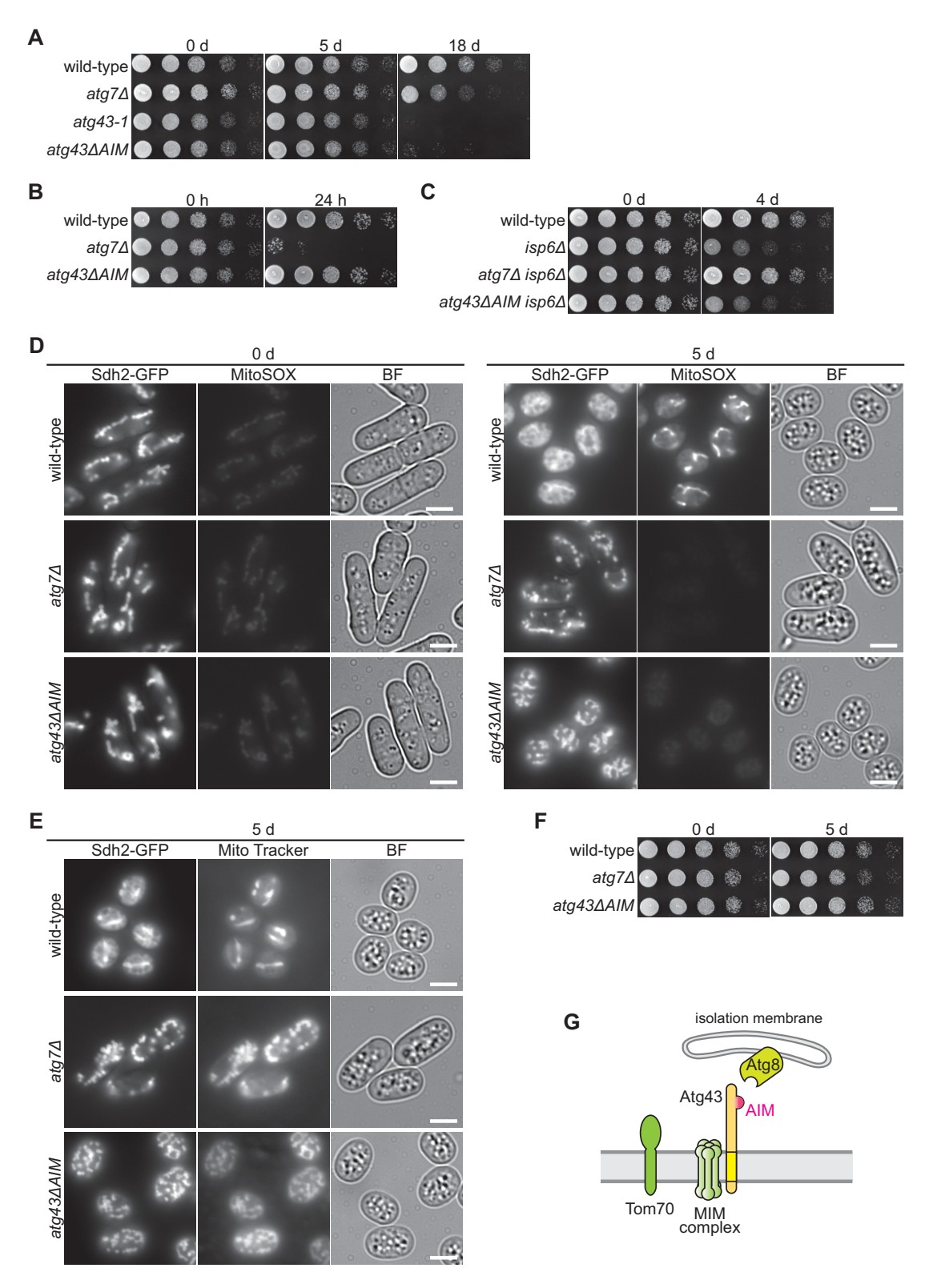

**Figure 7.** Mitophagy maintains growth ability during nitrogen starvation. (A–C) The indicated strains were grown in EMM and shifted to EMM–N. Cells were collected at the indicated time points after induction of nitrogen starvation and spotted onto solid YES medium for the growth assay. Cells auxotrophic for leucine, uracil, histidine, and lysine (*leu1-32, ura4, his7, lys1*) were used in (B). (D, E) Cells were collected at the indicated time points after induction of nitrogen starvation and stained with MitoSOX (D) or Mito Tracker (E). Sdh2-GFP was detected as a mitochondrial marker. Scale bars

*Figure 7 continued on next page*

*Figure 7 continued*

represent 5 µm. BF, bright-field image. (F) Serial dilutions of the indicated strains collected immediately before or 5 days after induction of nitrogen starvation were spotted onto glycerol plates. (G) Schematic model for mitophagy mediated by Atg43 on the MOM.

18 days (*Figure 7A*), raising possibilities that imbalance between cytosolic and mitochondrial protein stabilities has an impact on cellular integrity and that a cytosolic factor(s) that compensates the mitophagy defect is degraded by bulk autophagy.

To examine whether mitophagy contributes to redistribution of nutrient resources, we assessed cell viability of auxotrophic strains, which are highly sensitive to the autophagy-mediated nutrient supply (*Kohda et al., 2007*). After 24 hr of nitrogen starvation, viability was severely reduced in *atg7Δ* auxotrophic cells (*Figure 7B*). However, *atg43ΔAIM* cells remained viable to the same extent as wild-type cells (*Figure 7B*). We also used cells lacking the vacuolar protease Isp6, in which autophagy becomes toxic due to the accumulation of undegraded cargo in vacuoles (*Kohda et al., 2007*). We found that viability of *isp6Δ* cells during nitrogen starvation was restored by the *atg7Δ* mutation but not by *atg43ΔAIM* (*Figure 7C*). Taken together, these results suggest that the contribution of Atg43-mediated mitophagy to the nutrient supply and to the total amount of autophagosome formation is much smaller than that of bulk autophagy.

To explore the role of mitophagy in mitochondrial function, we stained mitochondria with Mito-SOX, a mitochondrial superoxide indicator. In the wild-type strain, the MitoSOX signal was much stronger in cells during nitrogen starvation than in vegetatively growing cells, indicating that superoxide is highly generated in mitochondria during starvation (*Figure 7D*). By contrast, the *atg7Δ* and *atg43ΔAIM* mutants did not exhibit such an increased in superoxide (*Figure 7D*). These results raise the possibility that oxidative phosphorylation in respiration is less active in cells lacking mitophagy during nitrogen starvation. However, the membrane potential is maintained in cells lacking mitophagy, as judged by staining with MitoTracker, a membrane potential-dependent dye (*Figure 7E*). Moreover, *atg7Δ* and *atg43ΔAIM* cells that underwent nitrogen starvation could grow on a respiratory medium that contains glycerol, a non-fermentable substrate, as a carbon source (*Figure 7F*), suggesting that loss of mitochondrial DNA or irreversible damage of mitochondria did not take place in those cells. Thus, the absence of mitophagy during nitrogen starvation brings about an alteration in mitochondrial oxidative status, although the underlying mechanism currently remains unknown.

Taken together, we propose that the Atg43-deficient mutant can be a useful model to investigate the cellular functions of mitophagy and that use of the combination of the Atg43-deficient mutant and an autophagy-defective mutant, such as *atg7Δ*, allows the distinction between defects derived from the loss of mitophagy and bulk autophagy.

## Discussion

Selective autophagy is a process that leads to the degradation of specific organelles and cytosolic components and has been studied primarily in budding yeast and mammalian cells (*Fukuda and Kanki, 2018*; *Gatica et al., 2018*; *Pickles et al., 2018*). In fission yeast, mitochondria and the ER are degraded by autophagy using the core autophagy machinery, as well as Atg20- and Atg24-family proteins (*Takeda et al., 2010*; *Zhao et al., 2016*). However, it is unclear whether this starvation-induced degradation process is selective. In the present study, we have identified that the MOM protein Atg43, along with mitochondrial import factors, serves as a receptor for mitophagy. These findings suggest that mitochondria undergo selective autophagy in fission yeast. Atg43 is the first mitophagy receptor identified in fission yeast, indicating that receptor-mediated selective autophagy is conserved in this model organism. Hence, our findings enable the identification of general and specific characteristics of mitophagy among different organisms.

We have shown that the minimal elements of Atg43 necessary for mitophagy are the AIM and mitochondrial localization, suggesting that the primary role of Atg43 in mitophagy is tethering Atg8 to the MOM. Consistent with these findings, the artificial loading of Atg8 to mitochondria replaces the function of Atg43 in mitophagy. This suggests that the developing isolation membrane is stabilized on mitochondria via the interaction between Atg43 and Atg8 for efficient engulfment (*Figure 7G*). We propose that Atg43 acts as a selective autophagy receptor and ensures this

selectivity by linking mitochondria to the isolation membrane. Although the known mitophagy receptors and Atg43 do not share sequence homology, they similarly contain the cytosol-exposed N-terminal AIM and C-terminal transmembrane domain(s) (*Kirkin and Rogov, 2019*). Atg43 appears to be functionally similar to mammalian mitophagy receptors rather than to Atg32 in budding yeast, in that interaction with Atg8 is essential for mitophagy. Further study of Atg43 will provide insights into the overall understanding of the mechanisms underlying receptor-mediated mitophagy in mammals.

Mitophagy requires mitochondrial import factors, the MIM complex and Tom70, which facilitate the sorting of mitochondrial proteins (*Baker et al., 2007*; *Dimmer et al., 2012*; *Stefan Dimmer and Rapaport, 2010*). Mitochondrial localization and protein expression levels of Atg43 are impaired in the absence of the MIM complex. This suggests that the MIM complex contributes to the integration of Atg43 to the MOM and that its absence results in degradation of Atg43 that may be mislocalized in the cytosol. Although the mitophagy defect in *tom70Δ* cells is suppressed by the forced loading of Atg43 to the MOM, Atg43 remains on mitochondria in the absence of Tom70. This may be because, without Tom70, mitochondrial Atg43 is not fully functional, possibly due to inappropriate topology, incorrect conformation, lack of activation, or loss of interaction with other proteins. Alternatively, Tom70 may promote mitophagy in parallel with Atg43 by interacting with the autophagy machinery or facilitating the mitochondrial import of proteins required for mitophagy. A recent report suggests that the MIM complex and Tom70 in budding yeast are involved in the MOM integration of Atg32 (*Vitali et al., 2020*). The roles of mitochondrial import factors in mitophagy will be revealed in organisms other than fission yeast.

Complete deletion of the *atg43*+ gene leads to a growth defect, suggesting that Atg43 also has a mitophagy-independent function. As the C-terminal truncated Atg43 defective in mitochondrial localization has a growth defect, the mitophagy-independent function is likely to be exerted on mitochondria. In this mitophagy-independent function, Atg43 appears to act with the MIM complex, as normal cell growth requires aa 165–184 of Atg43, which are required for interaction with the MIM complex. Atg43 may facilitate the integration of MOM proteins together with the MIM complex. However, if so, Atg43 may not necessarily be involved in all of the import events mediated by the MIM complex, as the mitochondrial loading of Tom70 is highly dependent on the MIM complex but not on Atg43. Alternatively, similar to pATOM36 of trypanosomes, a functional analog of the MIM complex, Atg43 and the MIM complex may contribute to mitochondrial DNA segregation (*Vitali et al., 2018*). Notably, the C-terminal component of Atg43, which is required for the interaction with the MIM complex and normal cell growth, is shared among fungi. This suggests that the mitophagy-independent function of Atg43 is likely conserved among species. Conversely, the N-terminal component of Atg43, which contains the AIM necessary for mitophagy, is not conserved outside of the *Schizosaccharomyces* species. We thus hypothesize that Atg43 originally played a mitophagy-independent role on the MOM and later acquired function as a mitophagy receptor through evolution. During evolution, each organism may have selected and exploited a variety of MOM proteins for use as mitophagy receptors.

Mitophagy is thought to play a key role in the quality and quantity control of mitochondria. Molecular components involved in mitophagy have been identified and characterized, but the physiological significance of mitophagy remains unclear. Especially in mammals, mitophagy is complex, as different mitophagy pathways include multiple receptors and adapters and operate in different cell types in response to a variety of stimuli (*Kirkin and Rogov, 2019*; *Pickles et al., 2018*). By contrast, Atg43-mediated mitophagy is the sole pathway that degrades mitochondria, at least upon nitrogen starvation. Therefore, the cellular functions of mitophagy can be easily investigated by using mitophagy-defective mutants of *atg43*+, such as the *atg43-1*, AIM-mutated, and N-terminal truncated mutants, as a model of mitophagy deficiency. In the present study, we have shown that lack of mitophagy during nitrogen starvation affects the oxidative status of mitochondria possibly due to the reduced respiration activity or the accumulation of superoxide scavengers or antioxidants in mitochondria. Detailed phenotypic analyses of the mitophagy-defective mutants will contribute to the understanding of the physiological roles of mitophagy. In summary, our findings suggest that Atg43 plays a role in mitophagy in fission yeast and establish fission yeast as a useful model to gain mechanistic, physiological, and evolutionary insights into mitophagy in eukaryotes.

# Materials and methods

**Key resources table**

| Reagent type (species) or resource | Designation | Source or reference | Identifiers | Additional information |
|---|---|---|---|---|
| Strain, strain background (*S. pombe*) | 972 | Laboratory stock | ATCC24843 | Stocked in T. Kanki lab. |
| Strain, strain background (*S. pombe*) | CA101 | Laboratory stock | | *h- leu1-32*<br>Stocked in T. Kanki lab. |
| Strain, strain background (*S. pombe*) | CA103 | Laboratory stock | | *h- ura4-D18*<br>Stocked in T. Kanki lab. |
| Strain, strain background (*S. pombe*) | MM72-1D | NBPR (https://yeast.nig.ac.jp/yeast/) | FY7652 | *h- leu1-32 ura4-D18*<br>Stocked in T. Kanki lab. |
| Strain, strain background (*S. pombe*) | TFSP407 | This study | | *h- ura4-D18 tuf1-mRFP::ura4*<br>Stocked in T. Kanki lab. |
| Strain, strain background (*S. pombe*) | TFSP1079 | This study | | *h- ura4-D18 tuf1-mRFP::ura4 atg43-1::kanMX4*<br>Stocked in T. Kanki lab. |
| Strain, strain background (*S. pombe*) | TFSP1669 | This study | | *h- ura4-D18 tuf1-mRFP::ura4 tom70Δ::nat*<br>Stocked in T. Kanki lab. |
| Strain, strain background (*S. pombe*) | TFSP669 | This study | | *h- ura4-D18 tuf1-mRFP::ura4 sdh2-mEGFP::nat*<br>Stocked in T. Kanki lab. |
| Strain, strain background (*S. pombe*) | TFSP3834 | This study | | *h- ura4-D18 tuf1-mRFP::ura4 sdh2-mEGFP::nat atg43-1::kanMX4*<br>Stocked in T. Kanki lab. |
| Strain, strain background (*S. pombe*) | TFSP1013 | This study | | *h- atg43-1::kanMX4*<br>Stocked in T. Kanki lab. |
| Strain, strain background (*S. pombe*) | TFSP1057 | This study | | *h- ura4-D18 tuf1-mRFP::ura4 atg43Δ::hph*<br>Stocked in T. Kanki lab. |
| Strain, strain background (*S. pombe*) | TFSP3353 | This study | | *h- ura4-D18 tuf1-mRFP::ura4 atg43-1::kanMX4 pNATZA31*<br>Stocked in T. Kanki lab. |
| Strain, strain background (*S. pombe*) | TFSP1941 | This study | | *h- ura4-D18 tuf1-mRFP::ura4 atg43-1::kanMX4 pNATZA31-atg43*<br>Stocked in T. Kanki lab. |
| Strain, strain background (*S. pombe*) | TFSP3473 | This study | | *h- atg43Δ::kan pNATZA31*<br>Stocked in T. Kanki lab. |
| Strain, strain background (*S. pombe*) | TFSP2705 | This study | | *h- atg43Δ::kan pNATZA31-atg43*<br>Stocked in T. Kanki lab. |
| Strain, strain background (*S. pombe*) | TFSP183 | This study | | *h- ura4-D18 GFP-atg8::ura4*<br>Stocked in T. Kanki lab. |
| Strain, strain background (*S. pombe*) | TFSP991 | This study | | *h- ura4-D18 GFP-atg8::ura4 atg43-1::kanMX4*<br>Stocked in T. Kanki lab. |
| Strain, strain background (*S. pombe*) | TFSP743 | This study | | *h- pgk1-mEGFP::hph*<br>Stocked in T. Kanki lab. |
| Strain, strain background (*S. pombe*) | TFSP989 | This study | | *h- pgk1-mEGFP::hph atg43-1::kanMX4*<br>Stocked in T. Kanki lab. |
| Strain, strain background (*S. pombe*) | TFSP671 | This study | | *h- ura4-D18 tuf1-mRFP::ura4 yop1-mEGFP::nat*<br>Stocked in T. Kanki lab. |

*Continued on next page*

*Continued*

| Reagent type (species) or resource | Designation | Source or reference | Identifiers | Additional information |
|---|---|---|---|---|
| Strain, strain background (*S. pombe*) | TFSP987 | This study | | h- ura4-D18 tuf1-mRFP::ura4 yop1-mEGFP::nat atg43-1::kanMX4 Stocked in T. Kanki lab. |
| Strain, strain background (*S. pombe*) | TFSP25 | This study | | h- sdh2-mEFGP::kan tuf1-mRFP::hph Stocked in T. Kanki lab. |
| Strain, strain background (*S. pombe*) | TFSP137 | This study | | h- sdh2-mEFGP::kan tuf1-mRFP::hph atg1Δ::nat Stocked in T. Kanki lab. |
| Strain, strain background (*S. pombe*) | TFSP145 | This study | | h- sdh2-mEFGP::kan tuf1-mRFP::hph isp6Δ::nat Stocked in T. Kanki lab. |
| Strain, strain background (*S. pombe*) | TFSP1273 | This study | | h- tom70-mEGFP::hph Stocked in T. Kanki lab. |
| Strain, strain background (*S. pombe*) | CA1258 | This study | | h- tuf1-mRFP::kan Stocked in T. Kanki lab. |
| Strain, strain background (*S. pombe*) | TFSP78 | This study | | h- tuf1-mRFP::kan fun14Δ::nat Stocked in T. Kanki lab. |
| Strain, strain background (*S. pombe*) | TFSP1321 | This study | | h- tom70-mEGFP::hph atg43-1::kanMX4 Stocked in T. Kanki lab. |
| Strain, strain background (*S. pombe*) | TFSP3475 | This study | | h- mic60-mEGFP::nat Stocked in T. Kanki lab. |
| Strain, strain background (*S. pombe*) | TFSP3477 | This study | | h- mic60-mEGFP:: nat atg43-1::kanMX4 Stocked in T. Kanki lab. |
| Strain, strain background (*S. pombe*) | TFSP209 | This study | | h- atg1Δ::nat Stocked in T. Kanki lab. |
| Strain, strain background (*S. pombe*) | TFSP271 | This study | | h- isp6Δ::hph Stocked in T. Kanki lab. |
| Strain, strain background (*S. pombe*) | TFSP667 | This study | | h- ura4-D18 tdh1-mRFP:: ura4 sdh2-mEGFP::nat Stocked in T. Kanki lab. |
| Strain, strain background (*S. pombe*) | TFSP985 | This study | | h- ura4-D18 tdh1-mRFP::ura4 sdh2-mEGFP::nat atg43-1::kanMX4 Stocked in T. Kanki lab. |
| Strain, strain background (*S. pombe*) | TFSP1677 | This study | | h- ura4-D18 GFP-atg8:: ura4 tom70Δ::nat Stocked in T. Kanki lab. |
| Strain, strain background (*S. pombe*) | TFSP1675 | This study | | h- pgk1-mEGFP::kan tom70Δ::nat Stocked in T. Kanki lab. |
| Strain, strain background (*S. pombe*) | TFSP1679 | This study | | h- ura4-D18 tuf1-mRFP::ura4 yop1-mEGFP::nat tom70Δ::hph Stocked in T. Kanki lab. |
| Strain, strain background (*S. pombe*) | CA9659 | DOI:10.1111/j.1365–2443.2006.01025.x | | h+ ura4-D18 tor2-13::ura4 |
| Strain, strain background (*S. pombe*) | TFSP1539 | This study | | h- atg7Δ::nat Stocked in T. Kanki lab. |
| Strain, strain background (*S. pombe*) | TFSP1071 | This study | | h- ura4-D18 tuf1-mRFP:: ura4 hph::P81nmt:GFP-atg43 Stocked in T. Kanki lab. |

*Continued*

| Reagent type (species) or resource | Designation | Source or reference | Identifiers | Additional information |
|---|---|---|---|---|
| Strain, strain background (*S. pombe*) | TFSP3237 | This study | | *h- ura4-D18 tuf1-mRFP::ura4 hph::P81nmt:GFP-atg43 atg7Δ::nat* Stocked in T. Kanki lab. |
| Strain, strain background (*S. pombe*) | TFSP3177 | This study | | *h- tom70-mEGFP:: hph mic60-FLAG:: kan tuf1-mRFP::nat* Stocked in T. Kanki lab. |
| Strain, strain background (*S. pombe*) | TFSP1097 | This study | | *h- hph::P3nmt1:FLAG-atg43* Stocked in T. Kanki lab. |
| Strain, strain background (*S. pombe*) | TFSP1717 | This study | | *h- kan::P3nmt1:atg43-FLAG::hph* Stocked in T. Kanki lab. |
| Strain, strain background (*S. pombe*) | TFSP2633 | This study | | *h- hph::P3nmt1: FLAG-atg43(C80)* Stocked in T. Kanki lab. |
| Strain, strain background (*S. pombe*) | TFSP3593 | This study | | *h- kan::P3nmt1:atg43 (C80)-FLAG::hph* Stocked in T. Kanki lab. |
| Strain, strain background (*S. pombe*) | CA12620 | This study | | *h+ ura4-D18 tor2-13:: ura4 tuf1-mRFP::kan* Stocked in T. Kanki lab. |
| Strain, strain background (*S. pombe*) | TFSP1399 | This study | | *h- ura4-D18 tuf1-mRFP:: ura4 hph::P3nmt1:atg43* Stocked in T. Kanki lab. |
| Strain, strain background (*S. pombe*) | TFSP1101 | This study | | *h- ura4-D18 tuf1-mRFP:: ura4 hph::P3nmt1:FLAG-atg43* Stocked in T. Kanki lab. |
| Strain, strain background (*S. pombe*) | TFSP3205 | This study | | *h- ura4-D18 tuf1-mRFP:: ura4 hph::P3nmt1: FLAG-atg43ΔN20* Stocked in T. Kanki lab. |
| Strain, strain background (*S. pombe*) | TFSP3317 | This study | | *h- ura4-D18 tuf1-mRFP:: ura4 hph::P3nmt1 :FLAG-atg43ΔN40* Stocked in T. Kanki lab. |
| Strain, strain background (*S. pombe*) | TFSP3651 | This study | | *h- ura4-D18 tuf1-mRFP:: ura4 atg43ΔAIM::nat* Stocked in T. Kanki lab. |
| Strain, strain background (*S. pombe*) | TFSP3979 | This study | | *h- hph::P3nmt1: FLAG-atg43 kan:: P3nmt1:GFP-atg8* Stocked in T. Kanki lab. |
| Strain, strain background (*S. pombe*) | TFSP3981 | This study | | *h- hph::P3nmt1: FLAG-atg43ΔAIM kan:: P3nmt1:GFP-atg8* Stocked in T. Kanki lab. |
| Strain, strain background (*S. pombe*) | TFSP2721 | This study | | *h- ura4-D18 tuf1-mRFP:: ura4 sdh2-mEGFP::hph* Stocked in T. Kanki lab. |
| Strain, strain background (*S. pombe*) | TFSP3291 | This study | | *h- ura4-D18 tuf1-mRFP::ura4 sdh2-mEGFP::hph atg43ΔN41-80::nat* Stocked in T. Kanki lab. |
| Strain, strain background (*S. pombe*) | TFSP3293 | This study | | *h- ura4-D18 tuf1-mRFP::ura4 sdh2-mEGFP::hph atg43ΔN81-120::nat* Stocked in T. Kanki lab. |

*Continued*

| Reagent type (species) or resource | Designation | Source or reference | Identifiers | Additional information |
|---|---|---|---|---|
| Strain, strain background (*S. pombe*) | TFSP3307 | This study | | h- ura4-D18 tuf1-mRFP::ura4 sdh2-mEGFP::hph atg43ΔN121-164::nat Stocked in T. Kanki lab. |
| Strain, strain background (*S. pombe*) | TFSP3289 | This study | | h- atg43ΔN164::nat Stocked in T. Kanki lab. |
| Strain, strain background (*S. pombe*) | TFSP3315 | This study | | h- atg43ΔN184::nat Stocked in T. Kanki lab. |
| Strain, strain background (*S. pombe*) | TFSP1107 | This study | | h- atg43Δ::kan Stocked in T. Kanki lab. |
| Strain, strain background (*S. pombe*) | TFSP3261 | This study | | h- ura4-D18 tuf1-mRFP::ura4 pNATZA31-atg43(C80) hph:: P81nmt1:GFP-atg43ΔN164 Stocked in T. Kanki lab. |
| Strain, strain background (*S. pombe*) | TFSP4041 | This study | | h- hph::P3nmt1:FLAG-atg43 pNATZA1-GFP Stocked in T. Kanki lab. |
| Strain, strain background (*S. pombe*) | TFSP3983 | This study | | h- hph::P3nmt1:FLAG-atg43 pNATZA1-GFP-atg8 Stocked in T. Kanki lab. |
| Strain, strain background (*S. pombe*) | TFSP4263 | This study | | h- hph::P3nmt1:FLAG-atg43 pNATZA1-GFP-atg11 Stocked in T. Kanki lab. |
| Strain, strain background (*S. pombe*) | TFSP4265 | This study | | h- hph::P3nmt1:FLAG-atg43 pNATZA1-atg11-GFP Stocked in T. Kanki lab. |
| Strain, strain background (*S. pombe*) | TFSP1461 | This study | | h- atg43-1::kanMX4-FLAG::hph Stocked in T. Kanki lab. |
| Strain, strain background (*S. pombe*) | TFSP1021 | This study | | h- atg43-FLAG::hph Stocked in T. Kanki lab. |
| Strain, strain background (*S. pombe*) | TFSP4289 | This study | | h- tuf1-mRFP::nat atg43-1:: kanMX4-FLAG::hph Stocked in T. Kanki lab. |
| Strain, strain background (*S. pombe*) | TFSP3263 | This study | | h- ura4-D18 tuf1-mRFP::ura4 pNATZA31-atg43(C80) hph:: P81nmt1:GFP-atg43ΔN184 Stocked in T. Kanki lab. |
| Strain, strain background (*S. pombe*) | TFSP3223 | This study | | h- ura4-D18 tuf1-mRFP::ura4 sdh2-mEGFP:: hph pNATZA31-atg43(C80) Stocked in T. Kanki lab. |
| Strain, strain background (*S. pombe*) | TFSP3241 | This study | | h- ura4-D18 tuf1-mRFP::ura4 sdh2-mEGFP::hph pNATZA31- atg43(C80) atg43ΔC20::kan Stocked in T. Kanki lab. |
| Strain, strain background (*S. pombe*) | TFSP3253 | This study | | h- ura4-D18 tuf1-mRFP::ura4 sdh2-mEGFP::hph pNATZA31- atg43(C80) atg43ΔC60::kan Stocked in T. Kanki lab. |
| Strain, strain background (*S. pombe*) | TFSP3305 | This study | | h- ura4-D18 tuf1-mRFP::ura4 sdh2-mEGFP::hph pNATZA31- atg43(C80) atg43ΔC61-80::kan Stocked in T. Kanki lab. |
| Strain, strain background (*S. pombe*) | TFSP2419 | This study | | h- atg43ΔC20::hph Stocked in T. Kanki lab. |
| Strain, strain background (*S. pombe*) | TFSP3287 | This study | | h- atg43ΔC60::nat Stocked in T. Kanki lab. |

*Continued on next page*

*Continued*

| Reagent type (species) or resource | Designation | Source or reference | Identifiers | Additional information |
|---|---|---|---|---|
| Strain, strain background (*S. pombe*) | TFSP3363 | This study | | h- atg43ΔC61-80::nat Stocked in T. Kanki lab. |
| Strain, strain background (*S. pombe*) | TFSP3225 | This study | | h- ura4-D18 tuf1-mRFP::ura4 hph::P81nmt:GFP-atg43 pNATZA31-atg43(C80) Stocked in T. Kanki lab. |
| Strain, strain background (*S. pombe*) | TFSP3255 | This study | | h- ura4-D18 tuf1-mRFP::ura4 hph::P81nmt:GFP-atg43ΔC20::kan pNATZA31-atg43(C80) Stocked in T. Kanki lab. |
| Strain, strain background (*S. pombe*) | TFSP3259 | This study | | h- ura4-D18 tuf1-mRFP::ura4 hph::P81nmt:GFP-atg43ΔC60::kan pNATZA31-atg43(C80) Stocked in T. Kanki lab. |
| Strain, strain background (*S. pombe*) | TFSP3343 | This study | | h- ura4-D18 tuf1-mRFP::ura4 hph::P81nmt:GFP-atg43ΔC61-80::kan pNATZA31-atg43(C80) Stocked in T. Kanki lab. |
| Strain, strain background (*S. pombe*) | TFSP3559 | This study | | h- leu1-32 ura4-D18 tuf1-mRFP::ura4 pNATZA31-atg43(C80)::leu1MX4 atg43ΔC60-mEGFP::hph Stocked in T. Kanki lab. |
| Strain, strain background (*S. pombe*) | TFSP3529 | This study | | h- leu1-32 ura4-D18 tuf1-mRFP::ura4 pNATZA31-atg43(C80)::leu1MX4 kan::P3nmt1:GBP-fis1 atg43ΔC60-mEGFP::hph Stocked in T. Kanki lab. |
| Strain, strain background (*S. pombe*) | TFSP3453 | This study | | h- leu1-32 ura4-D18 tuf1-mRFP::ura4 pNATZA31-atg43(C80)::leu1MX4 Stocked in T. Kanki lab. |
| Strain, strain background (*S. pombe*) | TFSP3557 | This study | | h- leu1-32 ura4-D18 tuf1-mRFP::ura4 pNATZA31-atg43(C80)::leu1MX4 kan::P3nmt1:GBP-fis1 atg43ΔC60::hph Stocked in T. Kanki lab. |
| Strain, strain background (*S. pombe*) | TFSP3803 | This study | | h- ura4-D18 tuf1-mRFP::ura4 atg43-1::kanMX4 hph:P3nmt1:GBP-fis1 pNATZA1-GFP Stocked in T. Kanki lab. |
| Strain, strain background (*S. pombe*) | TFSP3793 | This study | | h- ura4-D18 tuf1-mRFP::ura4 atg43-1::kanMX4 hph::P3nmt1:GBP-fis1pNATZA1-GFP-atg43(N21-40) Stocked in T. Kanki lab. |
| Strain, strain background (*S. pombe*) | TFSP3785 | This study | | h- ura4-D18 tuf1-mRFP::ura4 atg43-1::kanMX4 hph::P3nmt1:GBP-fis1 pNATZA1-GFP-atg8 Stocked in T. Kanki lab. |
| Strain, strain background (*S. pombe*) | TFSP3165 | This study | | h- kan::P3nmt1:GFP-mim2 Stocked in T. Kanki lab. |
| Strain, strain background (*S. pombe*) | TFSP3163 | This study | | h- kan::P3nmt1:GFP-mim2 hph::P3nmt1:FLAG-atg43 Stocked in T. Kanki lab. |
| Strain, strain background (*S. pombe*) | TFSP3215 | This study | | h- kan::P3nmt1:GFP-mim1 Stocked in T. Kanki lab. |

*Continued*

| Reagent type (species) or resource | Designation | Source or reference | Identifiers | Additional information |
|---|---|---|---|---|
| Strain, strain background (*S. pombe*) | TFSP3213 | This study | | h- kan::P3nmt1:GFP-mim1 hph::P3nmt1:FLAG-atg43 Stocked in T. Kanki lab. |
| Strain, strain background (*S. pombe*) | TFSP3247 | This study | | h- ura4-D18 tuf1-mRFP::ura4 hph::P81nmt:GFP-atg43 mim1Δ::nat Stocked in T. Kanki lab. |
| Strain, strain background (*S. pombe*) | TFSP3173 | This study | | h- ura4-D18 tuf1-mRFP::ura4 hph::P81nmt:GFP-atg43 mim2Δ::nat Stocked in T. Kanki lab. |
| Strain, strain background (*S. pombe*) | TFSP3283 | This study | | h- ura4-D18 tuf1-mRFP::ura4 mim1Δ::nat Stocked in T. Kanki lab. |
| Strain, strain background (*S. pombe*) | TFSP3271 | This study | | h- ura4-D18 tuf1-mRFP::ura4 mim2Δ::nat Stocked in T. Kanki lab. |
| Strain, strain background (*S. pombe*) | TFSP1779 | This study | | h- ura4-D18 tuf1-mRFP::ura4 hph::P81nmt:GFP-atg43 tom70Δ::nat Stocked in T. Kanki lab. |
| Strain, strain background (*S. pombe*) | TFSP3565 | This study | | h- leu1-32 ura4-D18 tuf1-mRFP::ura4 pNATZA31-atg43 (C80)::leu1MX4 kan::P3nmt1:GBP-fis1 atg43ΔC60-mEGFP::hph mim1Δ::nat Stocked in T. Kanki lab. |
| Strain, strain background (*S. pombe*) | TFSP3567 | This study | | h- leu1-32 ura4-D18 tuf1-mRFP::ura4 pNATZA31-atg43 (C80)::leu1MX4 kan::P3nmt1:GBP-fis1 atg43ΔC60-mEGFP::hph mim2Δ::nat Stocked in T. Kanki lab. |
| Strain, strain background (*S. pombe*) | TFSP3569 | This study | | h- leu1-32 ura4-D18 tuf1-mRFP::ura4 pNATZA31-atg43(C80)::leu1MX4 kan::P3nmt1:GBP-fis1 atg43ΔC60-mEGFP::hph tom70Δ::nat Stocked in T. Kanki lab. |
| Strain, strain background (*S. pombe*) | TFSP3217 | This study | | h- kan::P3nmt1:GFP-mim2 hph::P3nmt1:FLAG-mim1 Stocked in T. Kanki lab. |
| Strain, strain background (*S. pombe*) | TFSP3761 | This study | | h- mim1Δ::nat Stocked in T. Kanki lab. |
| Strain, strain background (*S. pombe*) | TFSP3763 | This study | | h- mim2Δ::nat Stocked in T. Kanki lab. |
| Strain, strain background (*S. pombe*) | TFSP3161 | This study | | h- ura4-D18 tuf1-mRFP::ura4 tom70-mEGFP::kan Stocked in T. Kanki lab. |
| Strain, strain background (*S. pombe*) | TFSP3459 | This study | | h- ura4-D18 tuf1-mRFP::ura4 tom70-mEGFP::kan mim1Δ::nat Stocked in T. Kanki lab. |
| Strain, strain background (*S. pombe*) | TFSP3455 | This study | | h- ura4-D18 tuf1-mRFP::ura4 tom70-mEGFP::kan mim2Δ::nat Stocked in T. Kanki lab. |

*Continued*

| Reagent type (species) or resource | Designation | Source or reference | Identifiers | Additional information |
|---|---|---|---|---|
| Strain, strain background (*S. pombe*) | TFSP3479 | This study | | h- ura4-D18 tuf1-mRFP::ura4 kan::P3nmt1:GFP-mim1 Stocked in T. Kanki lab. |
| Strain, strain background (*S. pombe*) | TFSP3523 | This study | | h- ura4-D18 tuf1-mRFP::ura4 kan::P3nmt1: GFP-mim1 atg43Δ::nat Stocked in T. Kanki lab. |
| Strain, strain background (*S. pombe*) | TFSP3563 | This study | | h- ura4-D18 tuf1-mRFP::ura4 kan::P3nmt1: GFP-mim1tom70Δ::nat Stocked in T. Kanki lab. |
| Strain, strain background (*S. pombe*) | TFSP3175 | This study | | h- ura4-D18 tuf1-mRFP:: ura4 kan::P3nmt1:GFP-mim2 Stocked in T. Kanki lab. |
| Strain, strain background (*S. pombe*) | TFSP3525 | This study | | h- ura4-D18 tuf1-mRFP::ura4 kan::P3nmt1: GFP-mim2 atg43Δ::nat Stocked in T. Kanki lab. |
| Strain, strain background (*S. pombe*) | TFSP3561 | This study | | h- ura4-D18 tuf1-mRFP::ura4 kan::P3nmt1: GFP-mim2 tom70Δ::nat Stocked in T. Kanki lab. |
| Strain, strain background (*S. pombe*) | TFSP3485 | This study | | h- kan::P3nmt1:GFP-mim2 pNATZA31-atg43(C80) hph:: P3nmt1:FLAG-atg43 Stocked in T. Kanki lab. |
| Strain, strain background (*S. pombe*) | TFSP3487 | This study | | h- kan::P3nmt1:GFP-mim2 pNATZA31-atg43(C80) hph:: P3nmt1:FLAG-atg43ΔN164 Stocked in T. Kanki lab. |
| Strain, strain background (*S. pombe*) | TFSP3489 | This study | | h- kan::P3nmt1:GFP-mim2 pNATZA31-atg43(C80) hph:: P3nmt1:FLAG-atg43ΔN184 Stocked in T. Kanki lab. |
| Strain, strain background (*S. pombe*) | TFSP3601 | This study | | h- leu1-32 hph::P3nmt1: GFP-mim2 pNATZA31-atg43(C80) leu1MX4::P3nmt1:FLAG-atg43ΔC61-80::kan Stocked in T. Kanki lab. |
| Strain, strain background (*S. pombe*) | TFSP3617 | This study | | h- leu1-32 hph::P3nmt1: GFP-mim2 pNATZA31-atg43(C80) leu1MX4::P3nmt1: FLAG-atg43ΔC20::kan Stocked in T. Kanki lab. |
| Strain, strain background (*S. pombe*) | TFSP3603 | This study | | h- leu1-32 hph::P3nmt1: GFP-mim2 pNATZA31-atg43 (C80) leu1MX4::P3nmt1: FLAG-atg43ΔC60::kan Stocked in T. Kanki lab. |
| Strain, strain background (*S. pombe*) | TFSP3201 | This study | | h- tom70-FLAG::hph kan:: P3nmt1:GFP-mim2 Stocked in T. Kanki lab. |
| Strain, strain background (*S. pombe*) | TFSP3233 | This study | | h- hph::P3nmt1:FLAG-atg43 tom70-mEGFP::kan Stocked in T. Kanki lab. |
| Strain, strain background (*S. pombe*) | TFSP3649 | This study | | h- atg43ΔAIM::nat Stocked in T. Kanki lab. |

*Continued on next page*

*Continued*

| Reagent type (species) or resource | Designation | Source or reference | Identifiers | Additional information |
|---|---|---|---|---|
| Strain, strain background (*S. pombe*) | MKY1 | NBPR (https://yeast.nig.ac.jp/yeast/) | FY7455 | *h+ leu1-32 ura4 his7 lys1* |
| Strain, strain background (*S. pombe*) | TFSP3481 | This study | | *h+ leu1-32 ura4 his7 lys1 atg7Δ::kan* Stocked in T. Kanki lab. |
| Strain, strain background (*S. pombe*) | TFSP4259 | This study | | *h+ leu1-32 ura4 his7 lys1 atg43ΔAIM::nat* Stocked in T. Kanki lab. |
| Strain, strain background (*S. pombe*) | TFSP1559 | This study | | *h- isp6Δ::hph atg7Δ::nat* Stocked in T. Kanki lab. |
| Strain, strain background (*S. pombe*) | TFSP4255 | This study | | *h- isp6Δ::hph atg43ΔAIM::nat* Stocked in T. Kanki lab. |
| Strain, strain background (*S. pombe*) | TFSP10 | This study | | *h- sdh2-mEFGP::kan* Stocked in T. Kanki lab. |
| Strain, strain background (*S. pombe*) | TFSP4271 | This study | | *h- sdh2-mEFGP::kan atg7Δ::nat* Stocked in T. Kanki lab. |
| Strain, strain background (*S. pombe*) | TFSP4273 | This study | | *h- sdh2-mEFGP::kan atg43ΔAIM::nat* Stocked in T. Kanki lab. |
| Antibody | anti-RFP (Rabbit polyclonal) | MBL | Cat# PM005, RRID:AB_591279 | WB (1:3000) |
| Antibody | anti-GFP (Mouse monoclonal) | Takara Bio | Cat# 632380, RRID:AB_10013427 | WB (1:10000) |
| Antibody | anti-histone H3 (Rabbit polyclonal) | Abcam | Cat# ab1791, RRID:AB_302613 | WB (1:10000) |
| Antibody | anti-FLAG (Mouse monoclonal) | Sigma | Cat# F1804, RRID:AB_262044 | WB (1:4000) |
| Antibody | anti-DDDDK (Rabbit polyclonal) | MBL | Cat# PM020, RRID:AB_591224 | WB (1:4000) |
| Antibody | anti-Atp2 (Rabbit polyclonal) | Abcam | Cat# ab128743, RRID:AB_2810299 | WB (1:2000) |
| Antibody | anti-Actin (Mouse monoclonal) | Abcam | Cat# ab8224, RRID:AB_449644 | WB (1:4000) |
| Antibody | anti-mouse IgG (Goat polyclonal, Peroxidase conjugated) | Merck Millipore | Cat# AP124P, RRID:AB_90456 | WB (1:10000) |
| Antibody | anti-rabbit IgG (Goat polyclonal, Peroxidase conjugated) | Jackson ImmunoResearch | Cat# 111-035-003, RRID:AB_2313567 | WB (1:10000) |
| Antibody | anti-guineapig IgG (Goat polyclonal, Peroxidase conjugated) | Jackson ImmunoResearch | Cat# 106-035-003, RRID:AB_2337402 | WB (1:10000) |
| Antibody | anti-FLAG beads | Sigma | Cat# M8823, RRID:AB_2637089 | |
| Antibody | anti-GFP beads | Chromotek | Cat# gtma, RRID:AB_2631358 | |
| Antibody | anti-Atg43 (Guineapig polyclonal) | This study | | WB (1:4000) Stocked in T. Kanki lab. |
| Recombinant DNA reagent | pFA6a-leu1MX4 (plasmid) | NBPR | FYP2892 | |

*Continued on next page*

*Continued*

| Reagent type (species) or resource | Designation | Source or reference | Identifiers | Additional information |
|---|---|---|---|---|
| Recombinant DNA reagent | pJQW#600 (plasmid) | NBPR | FYP3955 | |
| Recombinant DNA reagent | pNATZA1 (plasmid) | NBPR | FYP2874 | |
| Recombinant DNA reagent | pNATZA31 (plasmid) | NBPR | FYP2879 | |
| Recombinant DNA reagent | pNATZA31-atg43 (plasmid) | This study | | Stocked in T. Kanki lab. |
| Recombinant DNA reagent | pNATZA31-atg43(C80) (plasmid) | This study | | Stocked in T. Kanki lab. |
| Recombinant DNA reagent | pNATZA1-GFP (plasmid) | This study | | Stocked in T. Kanki lab. |
| Recombinant DNA reagent | pNATZA1-GFP-atg8 (plasmid) | This study | | Stocked in T. Kanki lab. |
| Recombinant DNA reagent | pNATZA1-GFP-atg11 (plasmid) | This study | | Stocked in T. Kanki lab. |
| Recombinant DNA reagent | pNATZA1-atg11-GFP (plasmid) | This study | | Stocked in T. Kanki lab. |
| Recombinant DNA reagent | pNATZA1-GFP-atg43(N21-40) (plasmid) | This study | | Stocked in T. Kanki lab. |
| Commercial assay or kit | EzWestLumi One | Atto | WSE-7110 | |
| Commercial assay or kit | EzWestLumi plus | Atto | WSE-7120 | |
| Commercial assay or kit | Clarity Max Western ECL Substrate | Bio-Rad | 1705062 | |
| Chemical compound, drug | MitoTracker Red CMXRos | Thermo Fisher Scientific | M7512 | (250 nM) |
| Chemical compound, drug | Mito SOX Red | Thermo Fisher Scientific | M36008 | (5 µM) |
| Software, algorithm | Image Lab | Bio-Rad | | |
| Software, algorithm | MetaMorph 7 | Molecular Devices | | |

## Fission yeast strains and general techniques

The *S. pombe* strains used in this study are listed in Key Resources Table. The standard growth media and genetic manipulations for *S. pombe* were conducted as described previously (*Bähler et al., 1998*; *Chia et al., 2017*). The *S. pombe* integration vectors (pNATZA1 and pNATZA31) were obtained from the National Bio-Resource Project (NBRP). Cells were grown in complete YES medium (0.5% yeast extract and 3% glucose, supplemented with 225 mg/L of adenine, uracil, leucine, and histidine) and synthetic minimal EMM medium (Formedium). For respiration medium, YE glycerol (0.5% yeast extract and 3% glycerol) was used. For nitrogen starvation experiments, cells exponentially growing in EMM were pelleted by centrifugation, washed, and resuspended in EMM lacking ammonium chloride (EMM–N) (Formedium). Cells were grown at 30°C unless otherwise indicated. At least two biological replicates were performed for each experiment.

## Genetic screen for mitophagy-defective mutants

The haploid *S. pombe* deletion mutant library was purchased from Bioneer. Each mutant was transformed with a fragment coding Tuf1-mRFP (*tuf1-mRFP::nat*) and selected for nourseothricin resistance. The transformants were grown in YES, shifted to EMM–N, cultured overnight, and immunoblotted for mRFP. To generate the *atg43-1* allele, the deletion cassette and flanking regions

were amplified from the Bioneer *atg43* deletion strain using the primer pair, 5′-GTA ATG GTG CTT GGT TAC CAG CAA C-3′ and 5′-GCT TGT ATT CCA AAA GCA CCA TGC TC-3′.

## Spot test

For spot test analysis, liquid cell cultures were adjusted to cell density of $OD_{600}$ 1.0. Serial dilutions of the cells were spotted onto solid YES or YE glycerol media. Images were captured using the Bio-Rad Chemi Doc XRS imaging system and Image Lab software (Bio-Rad).

## Yeast two-hybrid assay

Appropriate DNA fragments were cloned into pGBDU-C1 and pGAD-C1 vectors (*James et al., 1996*). The *S. cerevisiae* strain PJ69-4A (*James et al., 1996*) was used as a host and the transformants were spotted onto solid SD media lacking leucine and uracil for maintenance of the plasmids, and media lacking adenine for the interaction test.

## Immunoblotting

Cells were fixed in 10% trichloroacetic acid, and crude cell lysates were prepared by breaking with glass beads in a buffer containing 8 M urea, 5% SDS, 40 mM Tris-HCl (pH 6.8), 0.1 mM EDTA, and 2-mercaptoethanol. Proteins were separated on a polyacrylamide gel and transferred onto a polyvinylidene difluoride membrane (Merck-Millipore). Signals were detected with horseradish peroxidase-conjugated secondary antibodies and visualized by ECL (Bio-Rad or Atto). Image data were acquired using the Bio-Rad Chemi Doc XRS imaging system and Image Lab software (Bio-Rad).

## Cell fractionation

For cell fractionation, cells exponentially growing in YES were pelleted and resuspended in a buffer containing 0.1 M Tris-HCl (pH 9.3) and 10 mM dithiothreitol. After incubation at 30℃ for 30 min, cells were harvested and resuspended in sorbitol buffer (1.2 M sorbitol and 20 mM potassium phosphate buffer, pH 7.5). Spheroplasts were prepared by treatment with Zymolyase-100T (Nacalai Tesque) and lysing enzymes from *Trichoderma harzianum* (Sigma). Spheroplasts were pelleted, rinsed in sorbitol buffer, resuspended in homogenization buffer (0.6 M sorbitol, 20 mM HEPES-KOH, pH 7.4), and homogenized using a glass homogenizer. Cell debris was removed by centrifugation at 1000 *g* for 10 min, and the mitochondria-rich fraction was collected by centrifugation at 6500 *g* for 10 min.

## Immunoprecipitation

For immunoprecipitation, cells were collected by centrifugation and disrupted in lysis buffer (50 mM Tris-HCl [pH 7.4], 150 mM NaCl, 0.5% or 1% NP-40, 10% glycerol, 1 mM phenylmethylsulfonyl fluoride, and complete protease inhibitors [Roche]), with glass beads using Micro Smash MS-100 (Tomy). Anti-FLAG M2 beads (Sigma) and GFP-Trap beads (Chromotek) were used to immunoprecipitate FLAG- and GFP-tagged proteins, respectively.

## Microscopy

Fluorescence microscopic analysis was conducted using the IX73 microscopy system (Olympus). Images were captured with the cooled charge-coupled device camera EXi Blue (QImaging) and processed using MetaMorph software (Molecular Devices) and Adobe Photoshop (Adobe Systems). At least 200 cells were analyzed in each condition.

## Mass spectrometry

Proteins co-purified with the 80 aa C-terminal region of Atg43 tagged with FLAG were analyzed by mass spectrometry using a nano-flow LC (Eksigent nanoLC 415) coupled with tandem mass spectrometer (TripleTOF 5600+, Sciex). Cells grown in YES were subjected to cell fractionation. Anti-FLAG immunoprecipitation was conducted for the mitochondrial fraction. Immunoprecipitants were subjected to tube-gel digestion with trypsin after reduction and alkylation using dithiothreitol and iodoacetamide, respectively, and then subjected to mass spectrometric analysis. Protein identification was conducted using the Mascot search engine (version 2.6) against the reference database of *S. pombe* (strain 972/ATCC 24843, SwissProt, downloaded April 3, 2018).

## Acknowledgements

We thank K Shiozaki, M Uritani, and the National Bio-Resource Project (NBRP), Japan, for reagents.

## Additional information

### Funding

| Funder | Grant reference number | Author |
| --- | --- | --- |
| Japan Society for the Promotion of Science | 17K07330 | Tomoyuki Fukuda |
| Japan Society for the Promotion of Science | 20K06552 | Tomoyuki Fukuda |
| Takeda Science Foundation | | Tomoyuki Fukuda |
| Japan Society for the Promotion of Science | 18H04858 | Tomotake Kanki |
| Japan Society for the Promotion of Science | 19H05712 | Tomotake Kanki |
| Japan Society for the Promotion of Science | 19K22419 | Tomotake Kanki |
| Japan Agency for Medical Research and Development | 20gm6110013h0003 | Tomotake Kanki |

The funders had no role in study design, data collection and interpretation, or the decision to submit the work for publication.

### Author contributions

Tomoyuki Fukuda, Conceptualization, Supervision, Funding acquisition, Validation, Investigation, Visualization, Methodology, Writing - original draft; Yuki Ebi, Tetsu Saigusa, Kentaro Furukawa, Shun-ichi Yamashita, Keiichi Inoue, Daiki Kobayashi, Yutaka Yoshida, Investigation; Tomotake Kanki, Conceptualization, Supervision, Funding acquisition, Investigation, Writing - original draft

### Author ORCIDs

Tomoyuki Fukuda (iD) https://orcid.org/0000-0003-2069-7127
Tomotake Kanki (iD) https://orcid.org/0000-0001-9646-5379

### Decision letter and Author response

Decision letter https://doi.org/10.7554/eLife.61245.sa1
Author response https://doi.org/10.7554/eLife.61245.sa2

## Additional files

### Supplementary files

• Transparent reporting form

### Data availability

All data generated or analysed during this study are included in the manuscript and supporting files.

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
