## [Decision Letter]

**Acceptance summary:**

This study identified the mitochondrial outer membrane protein Atg43 as the first receptor for mitophagy (selective degradation of mitochondria by autophagy) in fission yeast. Atg43 triggers mitophagy upon nitrogen starvation by tethering mitochondria to forming autophagosomal membranes, and Atg43-mediated mitophagy is crucial for cell survival during prolonged nitrogen starvation. Thus, this study reveals the fundamental mechanism and physiological impact of mitophagy in fission yeast and also provides significant insights into the evolution of mitophagy in eukaryotes.

**Decision letter after peer review:**

Thank you for submitting your article "Atg43 tethers the site of autophagosome formation to mitochondria with the help of the mitochondrial import factors" for consideration by *eLife*. Your article has been reviewed by three peer reviewers, one of whom is a member of our Board of Reviewing Editors, and the evaluation has been overseen by Suzanne Pfeffer as the Senior Editor. The following individuals involved in review of your submission have agreed to reveal their identity: Noboru Mizushima (Reviewer #2); Jin-A Lee (Reviewer #3).

The reviewers have discussed the reviews with one another and the Reviewing Editor has drafted this decision to help you prepare a revised submission.

Summary:

*Schizosaccharomyces pombe* has served as an excellent model organism for our understanding of mechanisms underlying different cellular activities, but selective autophagy in this organism remains poorly understood. In the present study, Fukuda et al. identified the first mitophagy receptor Atg43 in *S. pombe*. The authors showed that Atg43 is localized to the mitochondrial outer membrane via the Mim complex, and triggers mitophagy under nitrogen starvation via interaction with Atg8. Because forced targeting of Atg8 to mitochondria could bypass mitophagy defects in *atg43* mutant cells, the authors propose that the role for Atg43 is to tether the site of autophagosome formation to mitochondria. Although Atg43 was found to be important for normal cell growth independently of its function as a mitophagy receptor, the authors used a specific allele of *atg43*, which does not affect cell growth under normal conditions, to suggest that mitophagy is crucial for survival of *S. pombe* during prolonged nitrogen starvation. Overall, this study reveals the fundamental mechanism and physiological impact of mitophagy in *S. pombe* and also provides significant insights into the evolution of mitophagy in eukaryotes. However, the authors should address the following issues to strengthen their conclusions or improve the manuscript.

Essential revisions:

1) The authors should reconsider the manuscript title to address the following issues:

i) *Schizosaccharomyces pombe* (or fission yeast) should be included.

ii) In this study, the authors only look at starvation-induced mitophagy, in which "starvation" would be the trigger. This could be the reason why only the interaction of Atg34 with Atg8, not with the autophagy initiating Atg1 complex, is required. Thus, it may be better to modify the title to indicate that Atg43 is important for starvation-induced mitophagy in *S. pombe*.

iii) Given the result that forced Atg8 targeting to mitochondria restored mitophagy in *atg43* mutant cells, the authors state that the role for Atg43 is to tether the site of autophagosome formation to mitochondria. However, this would be an overstatement. In addition, in our current view established by studies of other organisms, autophagy receptors organize the site of autophagosome formation on (or in the vicinity of) degradation targets, and then bind to Atg8 lipidated in forming/expanding phagophores to tether the membranes to the targets. Accordingly, the authors should rephrase the manuscript title (and similar statements in the text).

iv) The authors showed that the Mim complex is involved in Atg43 integration into the mitochondrial outer membrane. They also found that Tom70 is important for mitophagy, but how this protein contributes to mitophagy remains unknown. Therefore, "with the help of the mitochondrial import factors" in the manuscript title will be misleading, because it sounds like these import factors are involved in the mitophagy receptor function of Atg43.

2) It would be better if the authors could describe what the upstream signaling of mitophagy associated with Atg43 is. Is it nitrogen starvation rather than expression of Atg43? Can rapamycin induce mitophagy?

3) The authors should show whether the mutation at the AIM of Atg43 also impairs mitophagy and reduces cell viability under starvation, because Atg43 has a mitophagy-independent function important for cell growth and therefore the phenotype of *atg43-1* may not solely represent a defect in mitophagy. The AIM mutant would be more reliable as a mitophagy-specific mutant.

4) Although the authors state in the first line of the Abstract that mitophagy is important for mitochondrial function, it is not determined in *atg43*-deficient cells; *atg43-1* mutant cells show reduced viability during long-term starvation, but its cause is unclear. The authors should investigate whether any defects in mitochondrial function (respiratory activity, retention of mitochondrial DNA, etc.) occur in *atg43* mutants, including the AIM mutant, under growing and starvation conditions.

---

## [Author Response]

Essential revisions:1) The authors should reconsider the manuscript title to address the following issues:i) *Schizosaccharomyces pombe* (or fission yeast) should be included.ii) In this study, the authors only look at starvation-induced mitophagy, in which "starvation" would be the trigger. This could be the reason why only the interaction of Atg34 with Atg8, not with the autophagy initiating Atg1 complex, is required. Thus, it may be better to modify the title to indicate that Atg43 is important for starvation-induced mitophagy in *S. pombe*.iii) Given the result that forced Atg8 targeting to mitochondria restored mitophagy in atg43 mutant cells, the authors state that the role for Atg43 is to tether the site of autophagosome formation to mitochondria. However, this would be an overstatement. In addition, in our current view established by studies of other organisms, autophagy receptors organize the site of autophagosome formation on (or in the vicinity of) degradation targets, and then bind to Atg8 lipidated in forming/expanding phagophores to tether the membranes to the targets. Accordingly, the authors should rephrase the manuscript title (and similar statements in the text).iv) The authors showed that the Mim complex is involved in Atg43 integration into the mitochondrial outer membrane. They also found that Tom70 is important for mitophagy, but how this protein contributes to mitophagy remains unknown. Therefore, "with the help of the mitochondrial import factors" in the manuscript title will be misleading, because it sounds like these import factors are involved in the mitophagy receptor function of Atg43.

Following the above suggestion, the title of the paper has been changed to “Atg43 tethers isolation membranes to mitochondria to promote starvation-induced mitophagy in fission yeast”. In the original title and text, we described, “Atg43 tethers the site of autophagosome formation”, but now we are convinced that this was misleading as our data show that Atg43 binds to Atg8 but not to the Atg1 complex. Accordingly, we also have changed the text as well (Abstract and Discussion, second paragraph).

2) It would be better if the authors could describe what the upstream signaling of mitophagy associated with Atg43 is. Is it nitrogen starvation rather than expression of Atg43? Can rapamycin induce mitophagy?

Because fission yeast is not so sensitive to rapamycin as budding yeast (Takahara and Maeda, Genes Cells 17, 2012), we cannot induce mitophagy efficiently by rapamycin (not shown). We thus have used the temperature-sensitive mutant of the TOR kinase-coding gene *tor2* (*tor2-13*) to specifically inactivate TORC1. Upon TORC1 inactivation, Atg43 expression was up-regulated (Figure 2C) and mitophagy was induced (new Figure 2—figure supplement 2A). Therefore, TORC1 signaling negatively regulates Atg43 expression and mitophagy. By replacing the promoter, we also confirmed that Atg43 expression during vegetative growth does not induce mitophagy (Figure 1F and Figure 2—figure supplement 2B). Thus, expression of Atg43 is not sufficient to induce mitophagy, and inactivation of TORC1 (and probably following dephosphorylation of Atg1 and Atg13) is necessary. This is consistent with that artificial tethering of Atg8 does not induce mitophagy in growing cells (Figure 4H).

3) The authors should show whether the mutation at the AIM of Atg43 also impairs mitophagy and reduces cell viability under starvation, because Atg43 has a mitophagy-independent function important for cell growth and therefore the phenotype of atg43-1 may not solely represent a defect in mitophagy. The AIM mutant would be more reliable as a mitophagy-specific mutant.

In addition to *atg43-1*, we show that the *atg43ΔAIM* mutant displays mitophagy defect (Figure 3C) and reduced viability after long-term starvation (new Figure 7A). We also have replaced *atg43-1* by *atg43ΔAIM* for phenotypic analyses of mitophagy-defective cells (new Figure 7B-F).

4) Although the authors state in the first line of the Abstract that mitophagy is important for mitochondrial function, it is not determined in atg43-deficient cells; atg43-1 mutant cells show reduced viability during long-term starvation, but its cause is unclear. The authors should investigate whether any defects in mitochondrial function (respiratory activity, retention of mitochondrial DNA, etc.) occur in atg43 mutants, including the AIM mutant, under growing and starvation conditions.

According to the comment, we examined mitochondrial membrane potential by staining with MitoTracker and found that mitophagy-deficient cells maintain the wild-type level of membrane potential (new Figure 7E). We also examined mitochondrial DNA maintenance by assessing their growth on respiration medium, which contains non-fermentable glycerol as the main carbon source, and found that they grew normally (new Figure 7F). On the other hand, when stained with MitoSOX, a specific dye for mitochondrial superoxide, wild-type mitochondria gave rise to a strong signal whereas cells lacking mitophagy exhibited little or no signal (new Figure 7D). These results indicate that respiration during nitrogen starvation highly generates mitochondrial superoxide and raises a possibility that oxidative phosphorylation in respiration is less active in the absence of mitophagy. We have added these descriptions to the revised manuscript (subsection “Atg43-deficient cells can be used as a model of mitophagy-defective cells”, third paragraph). Further detailed analysis of respiration activity requires a metabolic analyzer and establishment of experimental procedures optimized for fission yeast especially under starvation conditions. Thus, we are planning to conduct such analysis in our future follow-up study.